# Active Retrosynthetic Planning Aware of Route Quality

**Luotian Yuan**[1]*, **Yemin Yu**[2,4]*, **Ying Wei**[3]†, **Yongwei Wang**[1,4]†, **Zhihua Wang**[4], **Fei Wu**[1,4]†

[1]Zhejiang University, [2]City University of Hong Kong, [3]Nanyang Technological University
[4]Shanghai Institute for Advanced Study of Zhejiang University
`3180105619@zju.edu.cn, yeminyu2-c@my.cityu.edu.hk, ying.wei@ntu.edu.sg`
`{yongwei.wang, zhihua.wang, feiwu}@zju.edu.cn`

## Abstract

Retrosynthetic planning is a sequential decision-making process of identifying synthetic routes from the available building block materials to reach a desired target molecule. Though existing planning approaches show promisingly high solving rates and route qualities, the trivial route quality evaluation via pre-trained forward reaction prediction models certainly falls short of real-world chemical practice. An alternative option is to annotate the actual quality of a route, such as yield, through chemical experiments or input from chemists, but this often leads to substantial query costs. In order to strike the balance between query costs and route quality evaluation, we propose an Active Retrosynthetic Planning (ARP) framework that remains compatible with the established retrosynthetic planners. On one hand, the proposed ARP trains an actor that decides whether to query the quality of a reaction; on the other hand, it resorts to a critic to estimate the value of a molecule with its preceding reaction quality as input. Those molecules with high reaction qualities are preferred to expand first. We apply our framework to different existing approaches on both the benchmark and an expert dataset and demonstrate that it outperforms the existing state-of-the-art approach by 6.2% in route quality while reducing the query cost by 12.8%. In addition, ARP consistently plans high-quality routes with either abundant or sparse annotations.

## 1 Introduction

Planning a retrosynthetic route is a central challenge in organic synthesis, requiring the break-down of a target molecule into available building block materials through a sequence of reactions. This process comprises two main components: (1) single-step reaction prediction predicting feasible reactions in a single step, (2) multi-step planning recursively selecting optimal molecules and reactions across multiple steps, where evaluating and ranking routes are of pivotal importance in shaping the planning policy. Previous efforts on multi-step planning have focused on quickly accessing building block materials in a limited number of single-step calls, resulting in an up to a $99.47\%$ success rate Xie et al. (2022) on specific benchmarks. Regrettably, this emphasis on evaluating search aspects overlooks the chemical practicability of planned routes, i.e., whether a route is quality-effective in practice. For example, there is a short but low-quality route in Fig. 1. The crux revolves around the definition of reaction quality, which is simply the predictive probability of a reaction according to a pre-trained forward single-step prediction model in approaches such as Retro* Chen et al. (2020) and GRASP Yu et al. (2022). This single-step model is trained to predict feasibility rather

---

*These authors contributed equally to this work.
†Corresponding authors.

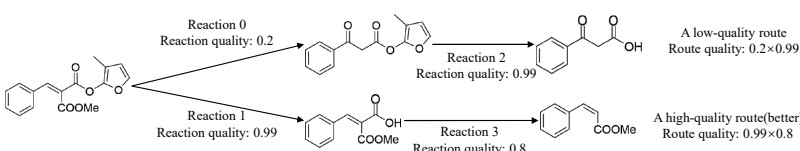

Figure 1: A short but low-quality route.

Figure 2: A low-quality route and a high-quality route. The low-quality reaction will lower the ceiling quality of the route. When selecting the next molecules towards high-quality routes, their preceding reaction qualities are associated consideration.

than the quality of reactions, thereby being biased towards highly feasible and frequent reactions instead of those with high qualities. An analysis illustrate this issue in Appendix B. The ideal reaction qualities that meet real-world chemical practicability, e.g., yield of a reaction, can be either annotated experimentally in a laboratory or by experienced chemists. Yet, annotating each reaction requires labor-intensive lab verification or expert annotations, compounded by the task of soliciting qualities for all reactions along a retrosynthetic route with route lengths ranging from 2.0 to 8.0(Yu et al. (2022),Liu et al. (2023a)). As verifying every reaction quality in the lab causes time delays and hinders automation, a quality metric is required which is expensive but not prohibitively so. In the real-life scenarios, such as online softwares like SYNTHIA(Lin et al. (2023)), it is an ideal candidate to integrate chemists into the AI planning process. Online annotations by chemists not only introduce minimal time delays and manageable labor costs, but also contribute valuable insights beyond mere reaction yields, such as toxicity, material costs, and work-up difficulty.

We are motivated to pursue a framework that strikes a balance between enhancing practical planning performance and minimizing annotation costs. The core idea of the proposed reinforcement learning-based Active Retrosynthetic Planning (ARP) framework constitutes an actor that decides whether to query the quality of a reaction or not and a critic that evaluates whether to expand a molecule or not. Concretely, the actor takes the current reaction as input; observing in Fig. 2 that a molecule with a high preceding reaction quality should be prioritized to expand first, the critic takes both the current molecule and its preceding reaction quality as input. It is noteworthy that the estimated molecule values by existing retrosynthetic planning methods can also be readily incorporated into our critic network. The critic network predicts a value that reflects the molecule's synthesizability together with the expectation of initializing a high-quality route. Simultaneously, the actor is trained to make the decision regarding the quality-effectiveness of querying for the annotated reaction qualities. Since querying for the reaction quality is a non-trivial task that induces an additional cost during annotation, we enforce a query cost to be paid whenever a query decision is made by the model. Given the diverse referential values of different reaction qualities during planning, we dynamically adjust the query cost to enhance the model's capability of identifying those reactions whose query results prove to be most quality-effective, thereby addressing the trade-off issue between query cost and the planning quality. The key contributions of our paper are outlined below. (1) *Practical efficacy:* we, for the first time, draw an insight into the disappointing practicality of existing retrosynthetic planners that regard single-step probabilities as reaction qualities. The ARP framework addresses the issue and improves quality of planned routes by 6.2% on a benchmark and by 4.9% on an annotated dataset. (2) *Generality:* The ARP framework is also compatible with arbitrary off-the-shelf planners and further boosts their chemical practicality.

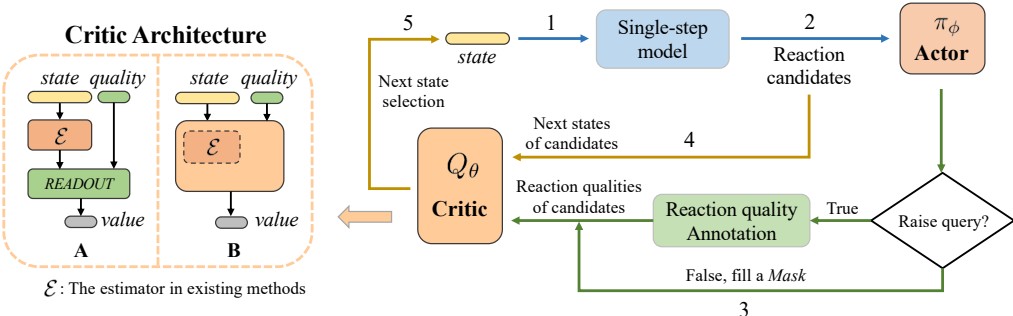

Figure 3: Our actor-critic framework. The actor receives the reaction candidates and outputs query decisions. The critic receives molecules along with preceding reaction qualities and outputs the estimated values. A pretrained offline estimator can be incorporated with the critic of architecture **A**. When training an online estimator with the critic simultaneously, ARP utilizes a simplified critic architecture variant **B**.

## 2   RELATED WORK

**Single-step prediction** Existing single-step methods can be divided into three main categories, template-based, template-free, and semi-template-based. Template-based methods pre-define reaction transformations as templates and select appropriate template candidates (Coley et al. (2017), Segler & Waller (2017), Dai et al. (2019),Chen & Jung (2021), Xie et al. (2023)). Template-free methods predict the reactants in the representation of SMILES sequences(Schwaller et al. (2019),Zhong et al. (2022)) or molecular graphs (Sacha et al. (2021)). Semi-template-based methods decompose the task into two stages, center identification, and synthon completion (Shi et al. (2020), Yan et al. (2020), Somnath et al. (2020)). Chen & Jung (2021) solves the identify-and-complete processes into a local template prediction through a global attention mechanism. Furthermore, Xie et al. (2023) address the issue that fxed parameters might be sub-optimal by the robust non-parametric local reaction template retrieval. Zhong et al. (2022) fixes the problem that SMILES neglects the characteristics of molecular graph topology and reaction atom transformations. Yu et al. (2023) sorts out two types of distribution shifts in retrosynthesis prediction.

**Multi-step planning** Lin et al. (2020) takes advantage of a single-step seq2seq model and Ishida et al. (2022) further introduces domain knowledge to guide the search direction. Chen et al. (2020) designs a neural-based $A^*$-like algorithm. Rather than a tree-search policy, Xie et al. (2022) proposes to combine a graph-based search policy with the traditional $A^*$ algorithm. Kim et al. (2021b) perform two self-improved iteration algorithms to imitate successful trajectories and find an optimal search policy. Afterward, Yu et al. (2022) is capable of biasing the retrosynthetic planning toward a favorable goal prescribed by chemists. Schreck et al. (2019) and Liu et al. (2023a) consider the cost for each reaction as a uniform 1 and optimize towards shortest routes. However, a short route might have a lower yield than a long route. Liu et al. (2023b) introduces a novel multi-step planning approach via in-context learning, departing from conventional search algorithms. However, the primary objective of Liu et al. (2023a) and Liu et al. (2023b) is still evaluating success rates of planned routes. They do not consider the real-world reaction qualities. Tripp et al. (2023), which shares a comparable motivation with ours, addresses the uncertainty of the stochastic retrosynthetic planning processes and focuses on finding several routes to complement their respective shortcomings. The methodologies above require the annotation of costs for every reaction along a route. In the pursuit of incorporating reliable reaction qualities derived from chemists or lab experimentation, which entail significant costs, these methods tend to be economically impractical for real-world deployment.

## 3 METHODS

### 3.1 ACTIVE RETROSYNTHETIC PLANNING MDP

In our work, we consider the active retrosynthetic planning scenario modeled as a Markov decision process (MDP), represented by $M = \{S, (A^r, A^q), \mathcal{P}, \mathcal{R}, c\}$. Specifically, $S$ denotes the state space comprising chemical molecules, $A^r$ refers to the action space of candidate reactions, and $A^q = \{0, 1\}$ represents the action space for query decisions. At time step $t$, given a molecule state $s_t$, the agent opts for a decision through a basic action pair $(a_t^r, a_t^q)$, where $a_t^r$ denotes a candidate reaction and $a_t^q$ indicates the agent's decision to query (or not) for the annotated reaction quality. The agent observes the reaction quality $u_t$ of $a_t^r$ only if $a_t^q = 1$. The $\mathcal{P}$ represents the deterministic state transition function from $s_t$ to $s_{t+1}$ via execution of reaction $a_t^r$. $\mathcal{R}$ denotes our reward function and $c$ is the constant query cost. Contrary to prior studies that employed a binary reward function (yielding a reward of 1 if the reaction reaches the building block materials, and 0 otherwise), this work associates the successful reward with both route quality and the query cost, as outlined in Eq. 1.

$$\mathcal{R}(s, a^r, a^q) = \begin{cases} \dfrac{\lambda + u}{\lambda + 1} - N_q c & if\ a^r\ \text{reach}\ I \\ 0 & \text{otherwise} \end{cases} \tag{1}$$

$I$ denotes the building block materials. $c^r$ represents the route quality, defined as the cumulative product of reaction qualities $\prod_{t=0} u_t$. Both route and reaction qualities fall within the interval $[0, 1]$. $N_q$ is the number of the annotated reactions, calculated by $\sum_{t=0} a_t^q$. $\lambda$ is a hyperparameter employed to stabilize the training when the agent obtains a success route with extremely small $u$.

### 3.2 MODEL FRAMEWORK AND TRAINING PROCEDURE

We employ an actor-critic framework for the approach. Given a reaction $a^r$, the actor $\pi_\phi\colon A^r \to A^q$ is responsible for making a query decision $a^q$, determining whether to query the reaction quality $u$ or not. The observation function $\mathcal{O}$ transforms the reaction and the query action to a $d^M$-dimensional embedding of the reaction qualities as expressed in Eq. 2. We use a binning strategy $\mathcal{B}$ to discretize the continuous reaction quality values into $N^M$ buckets and obtain the associated bucket embedding. The bins are constructed in order to cover a similar amount of reactions individually. In addition, we regard $\mathbb{M}$ as a separate bucket embedding. Details are given in Appendix C

$$\mathcal{O}(a^r, a^q) = \begin{cases} \mathcal{B}(u) & if\ a^q = 1 \\ \mathbb{M} & if\ a^q = 0 \end{cases} \tag{2}$$

For a given molecule $s$ with its corresponding reaction quality $u$, the critic $Q_\theta(s, u)$ estimates the value of $s$. The predicted values are used to access the value of the next state selection during RL roll-outs. To integrate existing retrosynthetic planners, a standard molecule estimator $\mathcal{E}\colon S \to \mathbb{R}$ is assumed within existing algorithms, whose task is to evaluate the values of molecules within their specific search frameworks. As summarized in Appendix A, existing methods can be roughly divided into offline and online categories. Offline methods, such as neural-based A* search in Chen et al. (2020) and Somnath et al. (2020), train the estimator on the extracted routes from the publicly available reaction databases in an offline manner, including the United States Patent Office (USPTO) or Pistachio(Mayfield et al. (2017)). In contrast, online methods such as Yu et al. (2022) leverage routes generated by online roll-outs. We design the critic network $Q_\theta(\mathcal{E}(s), u) = READOUT(\mathcal{E}(s), u)$ as architecture $A$ shown in Fig. 3. In this design, arbitrary pretrained offline planners can be seamlessly integrated into the critic. Additionally, online planners can be trained simultaneously with ARP training, simplifying the critic architecture to architecture $B$ in Fig. 3.

---

**Algorithm 1:** Training algorithm

---

**Initialize** the estimator $\mathcal{E}$, the actor policy $\pi_\phi$, the critic value function $Q_\theta$, the initial state $s_0$
**for** $t=0$ **to** $T$ **do**

> Observe reaction action space $\{a_i^r\}_{i=1}^k$ of $s_t$ from environment
> Make query decisions $\{a_i^q\}_{i=1}^k$ and observe reaction qualities $\{u_i = \mathcal{O}(a_i^r, a_i^q)\}_{i=1}^k$
> Observe the next step states $\{s_{t+1}^i\}_{i=1}^k$
> Select the next state $s_{t+1}$ by $\arg\max_{1 \le i \le k} Q_\theta(\mathcal{E}(s_{t+1}^i), u_i)$
> Compute reward $r_t = \mathcal{R}(s_t, a_t^r, a_t^q)$
> Append $(s_t, u_{t-1}, (a_t^r, a_t^q), r_t, s_{t+1})$ to the buffer

**end**
Update $\pi_\phi$ and $Q_\theta$ by Eq. 3 and Eq. 4

---

Such synergistic optimization not only conserves training resources for route collection but also substantially enhances both the solving rate and route quality compared in comparison to the two-stage training process.

At time step $t$, the agent observes state $s_t$ along with a batch of reaction candidates $\{a_i^r\}_{i=1}^k$. The actor chooses query actions $\{a_i^q\}_{i=1}^k$ and observes the corresponding reaction qualities $\{u_i = \mathcal{O}(a_i^r, a_i^q)\}_{i=1}^k$. Subsequently, the respective next states $\{s_{t+1}^i\}_{i=1}^k$ of reaction candidates are obtained by employing the state transition function $P$. The next state $s_{t+1}$ is then selected with the critic by $\arg\max_{1 \le i \le k} Q_\theta(\mathcal{E}(s_{t+1}^i), u_i)$. This comprises a complete step for executing a single step roll-outs, and a step reward $r_t = \mathcal{R}(s, a^r, a^q)$ is determined by the reward function in Eq. 1. The training procedure performs recursive roll-outs to collect trajectories for training and terminates each roll-out whenever it reaches dead/building-block molecules or maximum depth. We store the transition tuple $(s_t, u_{t-1}, (a_t^r, a_t^q), r_t, s_{t+1})$ into the replay buffer for training.

The actor-critic framework is trained using the TD3 algorithm. The target critic network is updated through the one-step TD equation where $Q'$ and $\pi'$ denote the target critic and actor networks, respectively, which is initialized using the same parameter from the main critic and actor networks $Q_\theta$ and $\pi_\phi$ but updated through an asynchronous manner. Utilizing the TD target $y_i$, the mean square error loss across the batch is computed for the original critic network $Q_\theta$ as follows:

$$y^{td} = r_t + \gamma Q'(\mathcal{E}(s_{t+1}), \mathcal{O}(a_t^r, \pi'(a_t^r))) \qquad L(\theta) = \frac{1}{N}\sum_i (y^{td} - Q_\theta(\mathcal{E}(s_t), u_{t-1})) \qquad (3)$$

Given the actor's objective is aligned toward maximizing the total return, and the critic network aims to approximate this cumulative return, the actor $\pi_\phi$ is trained by maximizing the Q-value, which is achieved through the minimization of loss in Eq. 4. If $\mathcal{E}$ is an offline estimator such as the value estimator in Retro*, it is trained before implementation in our framework. The parameters of $\mathcal{E}$ are frozen and not evolved in training of the critic and $\mathcal{E}' = \emptyset$. If $\mathcal{E}$ is an online estimator, it can be regarded as part of the critic parameters and is wrapped with the other critic parameters for training and $\mathcal{E}' = \mathcal{E}$. The algorithm is summarized in Algorithm 1.

$$L(\phi, \mathcal{E}') = -\frac{1}{N}\sum_i (-Q_\theta(\mathcal{E}(s_t), \mathcal{O}(a_t^r, \pi_\phi(a_t^r)))) \qquad (4)$$

### 3.3 INFERENCE PROCEDURE FOR ACTIVE RETROSYNTHETIC SEARCH

This section demonstrates the inference procedure for planning with partial observation of reaction qualities. In the inference stage, the reaction qualities are annotated by either a surrogate model or a chemist expert. We combine the actor $\pi_\theta$ and the critic $Q_\theta$ into the Monte-Carlo tree search (MCTS) on the AND-OR search

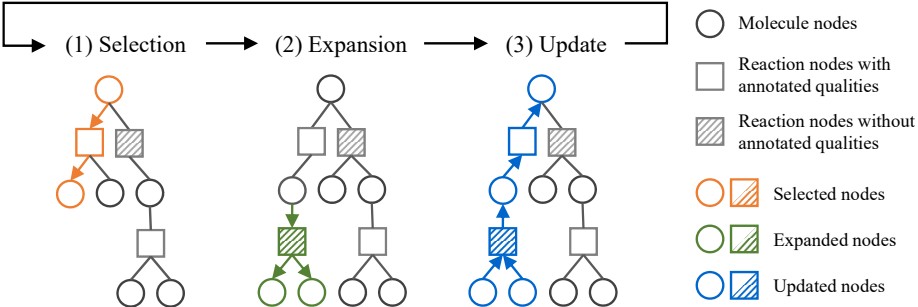

Figure 4: The active inference procedure. The active retrosynthetic search comprises three steps: selection, expansion, and update. Firstly, the planner selects a leaf molecule node with the maximum $Q^*$ value. Secondly, the selected node is expanded with an AND-OR stump, and $\pi_\phi(a^r)$ decides whether the new reaction node has an annotated reaction quality. Finally, update $Q^*$ of all nodes along the pathway.

tree. The molecule nodes are 'OR' nodes and the reaction nodes are 'AND' nodes. Moreover, we define a non-building block molecule node to be expandable when it is a leaf node or has an expandable child reaction node. None of the building block molecule nodes are expandable. A reaction node is expandable when there is any expandable child molecule node and none of the children are dead nodes. Molecule nodes are dead when they reach the maximum horizon or there are no reactions proposed by the single-step model. Let $ch(\cdot|T)$ denote the expandable children nodes of a molecule or reaction node.

There are three steps for each rollout during the active retrosynthetic search: selection, expansion, and update. A function $Q^*(s_t, a_t^r)$ in Eq. 5 presents the molecule synthesizability involves exploration. We use a UCT(Upper Confidence Bound applied to Trees) function(Kocsis et al. (2006)) to balance between exploitation of the route with the maximum $Q$ and exploration of those less frequently visited routes. $\beta$ is a hyper-parameter. $Q(s_t, a_t^r)$ is a value initialized by $Q_\theta(s_{t+1}, \mathcal{O}(a_t^r, \pi_\phi(a_t^r)))$ and updated afterward. A visit count number is denoted by $N(s, a^r)$ and initialized as 1.

$$Q^*(s_t, a_t^r) = Q(s_t, a_t^r) + \beta \frac{\sqrt{N(s_{t-1}, a_{t-1}^r)}}{1 + N(s_t, a_t^r)} \tag{5}$$

**Selection** The search tree $T$ starts with a single target root molecule node $s_0$. We recursively perform a reaction action selection $a_t^r = \arg\max_{a^r \in ch(s_t|T)} Q^*(s_t, a^r)$ and obtain the next state $s_{t+1}$ by tranition function $\mathcal{P}$ until reaching a leaf molecule node.

**Expansion** In the expansion step, we expand an AND-OR stump under the selected molecule node $s_t$ by referring to the single-step model. Every candidate reaction proposed is appended as a child reaction node of $s_t$. For a newly generated reaction node $a_t^r$, we assign its reaction quality observation by $\mathcal{O}(a_t^r, \pi_\phi(a_t^r))$. Each newly generated reaction node then expands children with the reactant molecule nodes.

**Update** During the update step, we perform $Q(s_t, a_t^r)$ values and visit counts $N(\cdot|T)$ backward traversal following the path from the selected leaf node back to the root node. $Q'$ is the $Q$ values of newly added state-action pairs. We use a simple moving average for the update with a discount factor $\gamma$ by Eq. 6.

$$Q'(s_t, a_t^r) = Q(s_t, a_t^r) + \frac{1}{N(s_t, a_t)}(\gamma Q' - Q(s_t, a_t^r)) \qquad N'(s_t, a_t^r) = N(s_t, a_t^r) + 1 \tag{6}$$

| | BENCHMARK TEST SET | | | EXPERT TEST SET | | |
|---|---|---|---|---|---|---|
| ALGORITHM | SUCCESS RATE ↑ | QUERY RATE ↓ | NORMALIZED ROUTE QUALITY ↑ | SUCCESS RATE ↑ | QUERY RATE ↓ | NORMALIZED ROUTE QUALITY ↑ |
| HGSEARCH | 36.5% | 100% | 0.518 | 49.0% | 100% | 0.550 |
| RETRO*-0 | 44.9% | 100% | 0.644 | 57.3% | 100% | 0.765 |
| RETRO* | 53.4% | 100% | 0.705 | 60.9% | 100% | 0.787 |
| RETRO*-0+ | 63.5% | 100% | 0.621 | 61.2% | 100% | 0.665 |
| RETRO*+ | 72.5% | 100% | 0.671 | 68.5% | 100% | 0.699 |
| EG-MCTS-0 | 73.6% | 100% | 0.489 | 65.4% | 100% | 0.726 |
| EG-MCTS | 85.4% | 100% | 0.558 | 75.4% | 100% | 0.712 |
| GRASP | 86.0% | 100% | 0.602 | 79.5% | 100% | 0.666 |
| RETRO*+ WITH ARP | 73.0% | 87.2% | **0.767** | 74.3% | 79.9% | 0.821 |
| GRASP WITH ARP | **86.5%** | 83.4% | 0.711 | **80.6%** | 90.8% | **0.836** |

Table 1: Performance of baselines and our approach on the benchmark and expert datasets.

# 4 EXPERIMENTS

## 4.1 EXPERIMENT SETUP

**Baselines.** We compare solving rate and route quality against various open-source baselines including a beam-search-like algorithm guided hyper-graph search method **HgSearch**(Schwaller et al. (2020)), best-first A*-like algorithm guided AND-OR tree search methods **Retro* and Retro*-0**(Chen et al. (2020)), and an experience-guided MCTS-based method **EG-MCTS**(Hong et al. (2023)). We apply ARP on an offline algorithm **Retro*+**(Kim et al. (2021a)) and an online one: **GRASP**(Yu et al. (2022)). **Retro*+** is based on **Retro*** with a self-improved single-step retrosynthetic model. **GRASP** applies a goal-driven actor-critic RL agent. As all of the existing methods rely on the reaction quality for calculating the search heuristics, we set their default query rate to 100%. We change the resource of reaction qualities from the single-step probabilities to our reaction qualities when testing the baselines.

**Materials and RL environment.** We use two test sets to evaluate our methods. The first one is a widely used USPTO-50k benchmark dataset that has 178 hard molecules raised in Chen et al. (2020). However, the scale of the benchmark dataset is small. Therefore, we include an additional expert dataset that has 8000 molecules and each molecule has a literature reference route extracted by chemist experts. The expert dataset is designed to emphasize retrosynthetic strategies with more challenging but strategically similar molecules. We partition the expert dataset as 0.8/0.1/0.1 into train/valid/test sets. The training set is used as the target molecules. For the single-step retrosynthesis model, we use a similar template-based model in Chen et al. (2020) which is a 2-layer MLP using the Morgan fingerprint as input. We adopt the top-50 single-step reaction candidates and set the maximum number of single-step inference calls as 100. We use the commercially available molecules dataset $eMolecules$ as the building block materials. As for the hyper-parameters, we set the maximum route depth as 6 and $\lambda$ in Eq. 1 as 4.

**Reaction quality annotation** Unfortunately, there is no established large-scale available reaction data set with respective reaction qualities or promising reaction performance, e.g. yield, prediction model(Jiang et al. (2022)), and both lab verification and expert annotations are expensive and time-consuming. We adopt a surrogate model to provide reaction quality annotations. Initially, the method in Guo et al. (2020) is employed to pre-train a model utilizing the USPTO-MIT dataset, followed by the fine-tuning of the model in reactions derived from the high-quality, expert-annotated dataset. Conceptually, the model, when trained on the expert-annotated dataset, prioritizes the identification of high-yield reactions over high-frequency reactions. More details about the training of surrogate model are in Appendix D. We provide an experiment to demonstrate a significant correlation between our surrogate model and reaction yields in Appendix E. Dur-

| ESTIMATOR | QUERY COST | ACTOR + CRITIC | | | RANDOM + CRITIC | | |
|---|---|---|---|---|---|---|---|
| | | SUCCESS RATE ↑ | QUERY RATE ↓ | NORMALIZED ROUTE QUALITY ↑ | SUCCESS RATE ↑ | RANDOM RATE | NORMALIZED ROUTE QUALITY ↑ |
| RETRO*+ | -0.01 | 72.4% | 100.0% | 0.772 | 72.4% | 100.0% | 0.772 |
| | 0 | 73.0% | 87.2% | 0.767 | 69.7% | 87.0% | 0.744 |
| | 0.01 | 72.4% | 63.6% | 0.759 | 71.3% | 64.0% | 0.722 |
| | 0.05 | 71.9% | 24.4% | 0.702 | 71.9% | 24.0% | 0.699 |
| | 0.1 | 73.5% | 0.0% | 0.685 | 72.6% | 0.0% | 0.685 |
| GRASP | 0 | 85.4% | 95.9% | 0.713 | 86.0% | 96.0% | 0.708 |
| | 0.005 | 86.5% | 83.4% | 0.711 | 84.8% | 83.0% | 0.695 |
| | 0.01 | 85.4% | 76.6% | 0.709 | 85.4% | 77.0% | 0.683 |
| | 0.02 | 86.0% | 13.4% | 0.681 | 86.0% | 13.0% | 0.666 |
| | 0.05 | 85.4% | 1.2% | 0.654 | 86.0% | 1.0% | 0.650 |

Table 2: Experimental results of the active query capability on the benchmark test set.

| ESTIMATOR | QUERY COST | ACTOR + CRITIC | | | RANDOM + CRITIC | | |
|---|---|---|---|---|---|---|---|
| | | SUCCESS RATE ↑ | QUERY RATE ↓ | NORMALIZED ROUTE QUALITY ↑ | SUCCESS RATE ↑ | RANDOM RATE | NORMALIZED ROUTE QUALITY ↑ |
| RETRO*+ | -0.01 | 73.1% | 100.0% | 0.830 | 73.1% | 100.0% | 0.830 |
| | 0 | 74.3% | 79.9% | 0.821 | 70.8% | 80.0% | 0.765 |
| | 0.01 | 71.2% | 53.0% | 0.790 | 72.6% | 53.0% | 0.739 |
| | 0.05 | 71.5% | 12.5% | 0.737 | 72.3% | 12.0% | 0.721 |
| | 0.1 | 74.5% | 0.0% | 0.722 | 73.1% | 0.0% | 0.722 |
| GRASP | 0 | 80.5% | 100.0% | 0.842 | 80.5% | 100.0% | 0.842 |
| | 0.005 | 80.6% | 90.8% | 0.836 | 79.5% | 91.0% | 0.772 |
| | 0.01 | 80.4% | 85.4% | 0.839 | 80.3% | 85.0% | 0.776 |
| | 0.02 | 79.8% | 6.6% | 0.749 | 80.9% | 6.0% | 0.733 |
| | 0.05 | 80.0% | 0.0% | 0.732 | 80.0% | 0.0% | 0.732 |

Table 3: Experimental results of the active query capability on the expert test set.

ing deployment, it is practical to replace the surrogate model with a chemist to provide online annotations, e.g. a coarse-grained quality rating from 0 to 10.

**Evaluation metrics.** We use three main metrics to comprehensively evaluate the performance of different search algorithms. 1. **Success rate**: The success rate is defined as the percentage of solved molecules in the entire test set. 2. **Query rate**: The query rate is defined as the percentage of reactions that are annotated with the reaction qualities in the inference stage. 3. **Normalized route quality:** Given a route, we compute the route quality by a cumulative product of reaction qualities. Both the reactions and route qualities range in $[0, 1]$ and a larger value refers to a higher quality, also a lower quality. We introduce the normalized route quality to evaluate the route quality performance. For each target molecule in the inference test set, we perform an exhaustive brute-force search in limited depth and acquire the maximum $u_{\max}$ and minimum route quality $u_{\min}$ for the success route. Especially, if there is only one successful route, we assume the single route quality as $u_{\min}$ and $u_{\max} = u_{\min} + 0.01$. We further define the normalized route quality as Eq. 7 of a route quality $u$ to eliminate the impact of different target molecules.

$$\text{Quality}_{\text{norm}} = \frac{u_{\max} - u}{u_{\max} - u_{\min}} \tag{7}$$

## 4.2 Results

**Comparision with baselines** The performance of all methods are presented in Table. 1. Concerning both the normalized route quality and the query rate metrics, our approach achieves the best performance on both datasets. The best existing method in the normalized route quality metric is identified as **Retro\***. In the benchmark dataset, our approach outperforms **Retro\*** by a margin of 6.2 %, while achieving a reduction in the query rate by 12.8%. Similarly, within the expert dataset, our approach outperforms **Retro\*** by 4.9 % and lowers the query rate by 9.2%. Regarding the success rate, both **Retro\*** and **GRASP** achieve a moderately higher success rate compared to the original results, primarily due to the influence of the preceding reaction quality as a potential molecular feature in predicting molecular synthesizability.

**Active query capability** We evaluate the performance of ARP for balancing the trade-off between the query costs and reaction qualities. In the context of the active query setting, the models are tested under diverse query cost settings. Intuitively, setting a high query cost is to simulate a sparse-annotated environment, compelling the planner to rely on less annotated reactions. Conversely, a near-zero query cost emulates an abundant-annotated setting. The experimental results on both the benchmark and expert datasets are listed in the left columns of Tab. 2 and Tab. 3, respectively. As we increase the query cost, we observe a decline in the query rate, which mildly impacts the success rate and normalized route quality. The phenomenon reflects that our approach is capable of actively selecting reactions that contribute most significantly to the reaction qualities in retrosynthetic planning. To further explore the active query capability of our approach, we conduct an ablation study where the planner chooses to query the reaction quality of a reaction with a fixed random rate $p$ as Eq. 8 instead of employing the trained actor-network for making query decisions.

$$a_i^q \leftarrow \begin{cases} 1 & p \\ 0 & 1 - p \end{cases} \tag{8}$$

We adjust the fixed probability $p$ as the same query rate obtained from the baseline result under varied query cost settings in order to eliminate the confounding effect of different query numbers. The results are listed in the right columns of Tab. 2 and Tab. 3 and demonstrate that the actor adeptly selects the most informative reactions to enhance the planning performance. As the random rate $p$ escalates, the critic can utilize more annotated reactions, improving the precision of value estimation and, thereby, optimizing the search process toward discovering routes with higher quality. We observe that the success rate does not increase monotonically with the query rate. Optimizing the success rate and the route quality together can lead to certain trade-offs, as demonstrated by a case study in Appendix F. Additionally, the actor is specifically trained to identify and query the most informative reactions in order to achieve a higher route quality. As a result, when the trade-off appears between the success rate and the route quality, the actor+critic approach might not improve but suppress the success rate when compared to the random+critic baseline.

## 5 Conclusion

The paper proposed ARP, a novel retrosynthetic planning framework aware of the route quality. Unlike existing approaches using a labor-free but trivial reaction evaluation which is biased to the high-frequent reactions, ARP adopts a route-quality evaluation approach aware of chemical practicability Moreover, there exists a trade-off between enhancing the planning performance and saving the query costs of acquiring reaction and is able to perform an active selection of the most informative reactions to observe their reaction qualities. Experimental results demonstrate ARP's capability of capturing high-quality routes under either abundant or sparse-annotation environments.

## 6 ACKNOWLEDGEMENT

This work is funded by National Natural Scientific Foundation of China (No. 62037001), the Research Matching Grant Scheme (No. 9229082), the Starry Night Science Fund at Shanghai Institute for Advanced Study (Zhejiang University) and Shanghai AI Laboratory.

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

## A  EXISTING PLANNING ALGORITHMS

|  | 3N-MCTS | HgSearch | DFPN-E | RetroGNN | Metro | FusionRetro | Retro-fallback |
|---|---|---|---|---|---|---|---|
| Algorithm | Online | Offline | Offline | Offline | Online | Offline | Online |
|  | **EG-MCTS** | **Retro**$^*$ | **RetroGraph** | **GNN-Retro** | **SimulatedExp** | **GRASP** | **PDVN** |
| Algorithm | Online | Offline | Offline | Offline | Online | Online | Online |

Table 4: Existing online and offline retrosynthetic planning methods.

**Active reinforcement learning** An active reinforcement learning(ARL) agent learns when to pay query costs and observe rewards(Daniel et al. (2014)) or other signals. A wide range of work has focused on ameliorating the problem of defining a complete reward function on trajectories in complicated real-world tasks, i.e. automated driving and robot grasping(Christiano et al. (2023), Saunders et al. (2018), Subramanian et al. (2016), Daniel et al. (2014)). To minimize reliance on human experts, Krueger et al. (2020), Bellinger et al. (2020), and Schulze & Evans (2018) study the active measure reinforcement learning(AMRL) framework under multi-armed bandit and tabular settings. Furthermore, Warnell et al. (2018) and Knox & Stone (2009) propose the TAMER framework which takes into account the time delays and noise when the human, a "teacher", provides rewards online to the agent, a "student".

## B  SINGLE STEP PROBABILITY

As the single-step model is trained to predict feasible reactant precursors, it is biased towards frequent reactions instead of those with high qualities. We verify the issue that frequently collected reactions in a single-step dataset are not necessarily high-yield, which we substantiate based on an analysis from Schwaller et al. (2021) that explores yields reported in the open-source USPTO dataset.

The USPTO dataset with reaction yields in sub-gram scale(Schwaller et al. (2021)) contains a large number of reactions and a broad range of superclasses, and a reaction distribution closely resembling that of the USPTO single-step dataset, such as USPTO-MIT. The actual reaction yield distribution of the above dataset, originally presented in Schwaller et al. (2021), is depicted in Fig 5c. Notably, a significant proportion of reactions within the dataset exhibits relatively low yields, affirming that the USPTO single-step dataset is not inherently biased to high-yield reactions. Fig 5a (originally presented in Schwaller et al. (2021)) shows various superclasses of reactions, where each color corresponds to a superclass and the coverage area of each color roughly represents the frequency of that superclass of reactions in the dataset. Combining Fig 5a and Fig 5b, we conclude that high-frequency superclasses do not show a significant correlation with high yields. For example, the superclasses annotated in purple and cyan demonstrate low yields, with only the green reaction superclass corresponding to high yields in Fig 5b.

In summary, frequently collected reactions in a single-step dataset are not inevitable to be high-yield ones and the single-step probabilities are not biased to high-quality but high-frequency reactions.

## C  BINING STRATEGY

A bining strategy $\mathcal{B}$ is performed to discretize the continuous reaction quality values in and obtain the associated bucket embedding. The preceding reaction cost is concatenated in the format its representation as $\mathcal{O}(a^r, a^q)$ in Eq 2. Concretely, when $a^q = 1$, we derive $\mathcal{O}(a^r, a^q)$ from (1) discretizing continuous reaction quality values into $N^M$ discrete buckets, (2) learning $N^M$ trainable embeddings in $d^M$ dimensions for all

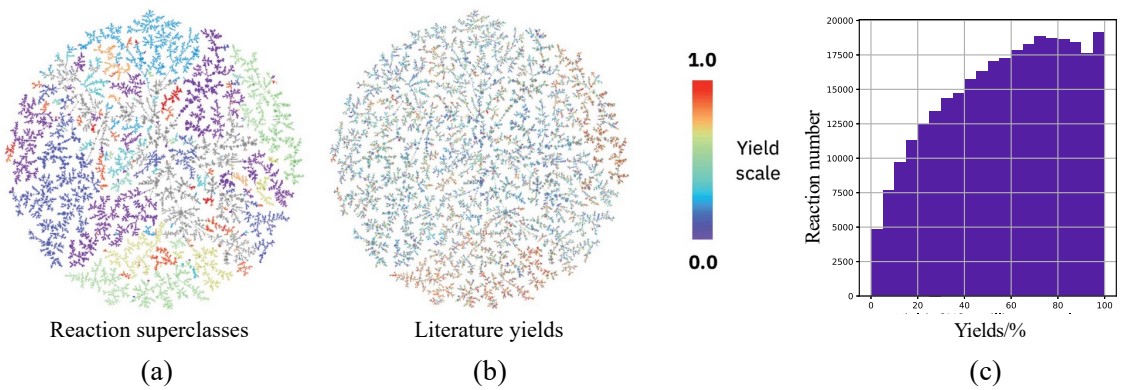

Reaction superclasses

(a)

Literature yields

(b)

(c)

Figure 5: The figure is directly borrowed from Schwaller et al. (2021). USPTO yield analysis: (a) shows the superclasses which roughly reflect the reaction frequency in the dataset. (b) depicts the yield scales of reactions labeled by the superclasses in (a). and (c) displays the distribution of the reaction yields in the dataset.

buckets within our critic, and (3) determining the bucket index that the queried quality value $u$ belongs to and thereby the associated bucket embedding. When $a^q = 0$, $\mathcal{O}(a^r, a^q) = \mathbb{M}$ as a $d^M$-dimensional trainable embedding. In our implementation, we consider $d^M = 512$ and $N^M = 18$ buckets which are defined in Fig 6 in the revision. These buckets are obtained via (1) collecting about 28M reactions during planning by GRASP and Retro*, (2) computing their reaction qualities by our surrogate model, and (3) defining the bucket boundaries to ensure that each bin covers a similar number of reactions.

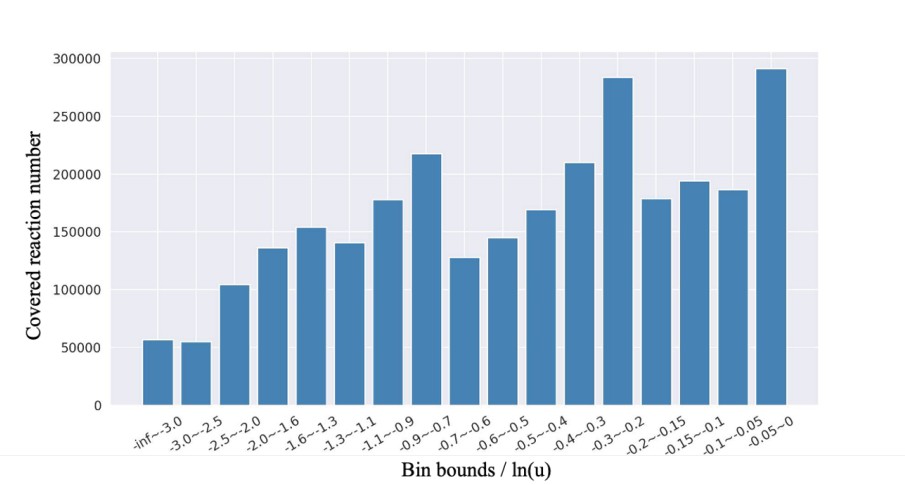

Figure 6: Bin bucket boundaries. Each bin covers a similar amount of collected reactions individually.

## D    Surrogate training details

We utilize a 8-layer Transformer as the architecture of our surrogate model. The hyper-parameters are listed in Tab 5. The training of our surrogate model involves two steps: (1) pre-training on the USPTO-MIT dataset, and (2) finetuning on an in-house expert dataset of routes featuring high-yield reactions. It is important to note that we introduce step (2) precisely to ensure that high predictive probabilities from our surrogate model align with high yields.

| Hyperparameters | Values |
|---|---|
| Encoder layers | 4 |
| Decoder layers | 4 |
| Encoder embedding dimension | 2048 |
| Encoder FFN embedding dimension | 2048 |
| Encoder attention heads | 8 |
| Decoder embedding dimension | 2048 |
| Decoder FFN embedding dimension | 2048 |
| Decoder attention heads | 8 |
| Optimizer | Adam |
| Learning rate | 1e-4 |
| Weight decay | 0.0001 |
| N epochs | 12 |
| Clip norm | 0.25 |
| Dropout rate | 0.1 |

Table 5: The output of the cross-validation used for the hyperparameters optimization

## E    Correlation between the surrogate model and reaction yields

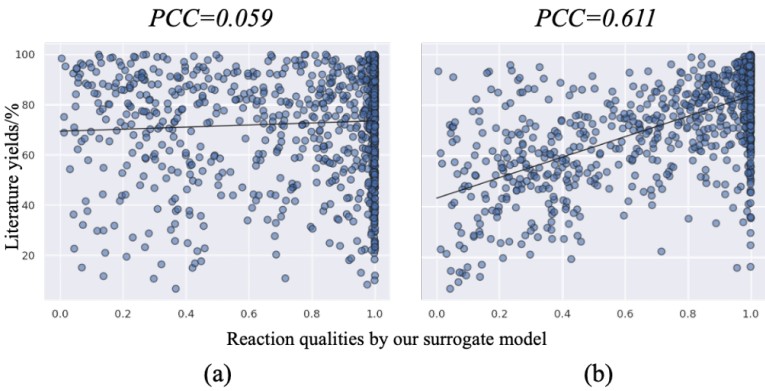

Figure 7: PCC against reaction yields. (a) shows the PCC of the pre-trained model and (b) shows the PCC of the finetuned model.

To evaluate our surrogate model, we resort to a route-with-yield test set. Following the method described in Chen et al. (2020), we extract synthesis routes with yields from the USPTO-milligram-scale reaction yield dataset Schwaller et al. (2021). For evaluation purposes, we randomly select 200 routes, encompassing approximately 1000 reactions. We thereby calculate the Pearson correlation coefficient(PCC) between the reaction quality predicted by our surrogate model and the literature yield. In Fig 7 of the revised manuscript, (a) illustrates the 0.059 PCC of the pre-trained model while (b) shows the 0.611 PCC of the finetuned model, providing strong evidence that the surrogate model accurately predicts yields.

## F SUCCESS RATE AND ROUTE QUALITY INCONSISTENCY

There are two separate objectives to optimize in our framework: the success rate and the route quality. However, optimizing these two objectives together can lead to certain trade-offs, as demonstrated by the following case study.

As shown in Fig. 8, the root molecule $M_0$ has three candidate reactions, and the $R_0$ is identified as a high quality reaction. However, the child molecule $M_1$ of $R_0$ is a unexpandable dead node with no further reaction candidates. If the planner makes the selection with observable next state molecular structures of $M_1$, $M_2$ and $M_3$ and unobservable reaction quality values of $R_0$, $R_1$ and $R_2$, it might selects $R_2$ for its most synthesizable next state molecule $M_3$. However, with observable reaction quality values, the planner could be misled into selecting $R_0$ due to its highest route quality expectation,which demonstrate an inconsistency between the two optimization obejectives.

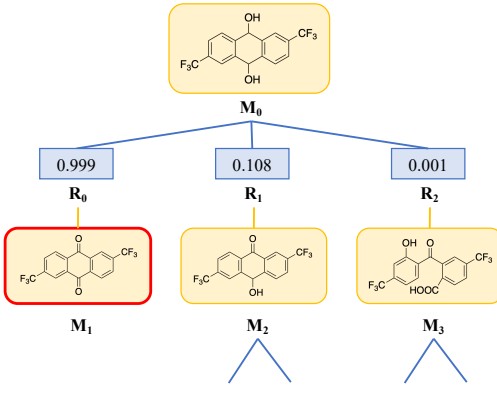

Figure 8: A case for illustrating two objective inconsistency. The root molecule $M_0$ has three candidate reactions, and the $R_0$ is identified as a high quality reaction. However, the child molecule $M_1$ of $R_0$ is a unexpandable dead node with no further reaction candidates.

## G REAL-LIFE RETROSYNTHETIC PLANNING SCENARIOS

The quality metric required by our framework in a real-life scenario should be expensive but not prohibitively so. While a single-step model is not competent enough, a lab validation might be excessively expensive and time-consuming. This consideration constitutes the primary motivation behind our active planning framework, aiming to query a minimum number of reaction quality annotations while still planing high-quality routes.

While our current implementation involves querying the surrogate model, our inspiration is drawn directly from real-life retrosynthesis planning scenarios, such as in online softwares like SYNTHIA, where chemists are pivotal end users. In this context, integrating chemists as valuable resources into the AI planning process will be invaluable for planning routes that are not only feasible but also of practical high-quality. We envision the successful deployment of our framework in this scenario for several reasons.

Online annotation by chemists introduces minimal time delays and manageable labor costs, making it an ideal candidate for a route quality metric that is expensive but not prohibitively so. Our framework is intentionally designed to be compatible with various types of annotations, including a coarse-grained quality rating from 0 to 10. We believe such a rating is sufficient for the planner to make satisfactory decisions. Additionally, this rating can also be seamlessly integrated into our current framework by replacing the bucket index to which a quality value belongs (see details in Section 3.2 in the revision) with this discrete rating. Chemists contribute valuable insights beyond mere reaction yields, such as knowledge about preferred reactions in real-world synthesis contexts, which can include factors like toxicity, material costs and work-up difficulty (post-process, like purification or separation).

## H  FUTURE WORK

Although we focused on the high-quality routes, the retrosynthetic planning has other essential considerations like the green chemistry. In future work, we intend to investigate Active Retrosynthetic Planning with multi-objective optimization in order to find eco-friendly routes of high chemical feasibility.

## I  CASE STUDY

We conduct a double-blind test to check the route quality generated by **Retro**, ARP with **Retro***, **GRASP**, and ARP with **GRASP**. We collect top-1 successful routes from the experimental results of the benchmark dataset and the chemists tag the route with a rating from 0 to 10. 10 refers to a high-quality route while 0 refers to a low-quality one. We lie the average rating in Tab. 6. Compared with the original methods, ARP with **Retro*** outperforms **Retro*** by 1.7 and ARP with **GRASP** outperforms **GRASP** by 2.2.

|  | **Retro*** | ARP with **Retro*** | **GRASP** | ARP with **GRASP** |
|---|---|---|---|---|
| Route rating(1-10) | 7.8 | 8.5 | 6.9 | 9.1 |

Table 6: Double blind test on the top-1 route quality.

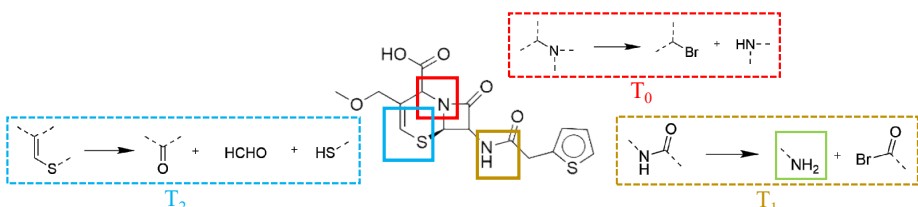

Figure 9: A target molecule.

Furthermore, we study a case to illustrate the active query capability. In Fig. 9, a target molecule has three basic molecular structures that need to be broken down by respective templates, $T_0, T_1, T_2$. Simplified,

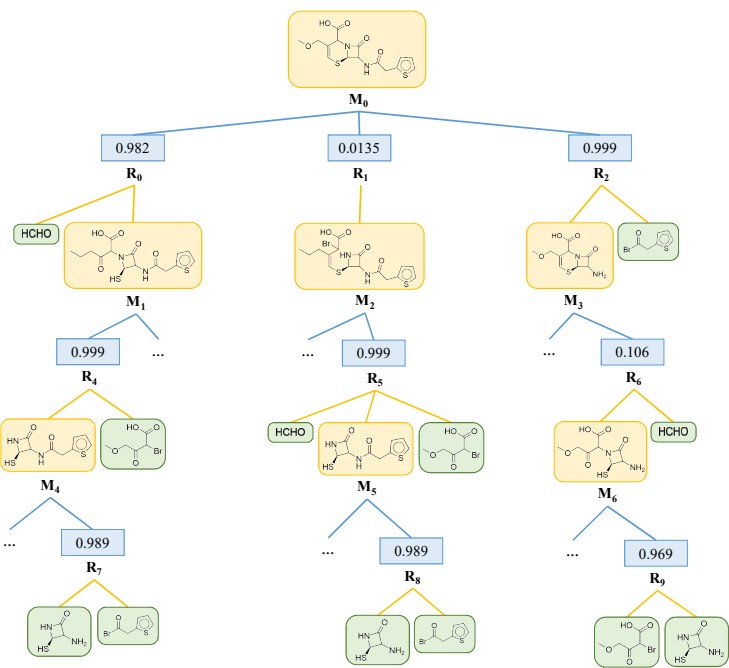

Figure 10: A search tree by ARP based on **GRASP**.

the planner needs to decide the order of executing three templates. However, if $T_0$ is executed after $T_1$, it will produce a low-quality reaction because $T_1$ reveals a high-activity amino group blocked green. From a chemical perspective, $T_1$ can be regarded as a deprotection reaction to suppress side reactions on the amino group for $T_0$. Thus $T_0$ must precede $T_1$. We visualize a search tree in Fig. 10 planned by ARP based on **GRASP** to solve the target molecule with the query cost equals 0, 0.01, and 0.05. For simplicity, we ignore some molecule nodes and reaction nodes. We tag the reaction qualities on the blue reaction nodes, the non-building block molecules on the yellow nodes, and the building block molecules on the green nodes. The empty blue nodes present reaction nodes of which the qualities are not annotated. Furthermore, we tag the $Q$ value near the respective molecule nodes to explore the reaction quality annotation's impact. In the three search trees, the molecule node selection among $M_0$, $M_1$, and $M_2$ is a key decision that determines the next following expansion of the whole search tree. $M_0$ will results in a high-quality route while $M_1$ and $M_2$ will lead to low-quality routes. $M_1$ has a low preceding reaction quality and $M_2$ has a low future-quality expectation. $M_1$ is the best next molecule node to expand. We compare the situations with different query costs.

**Ful observation:** With a query cost of 0, the actor in ARP queries every reaction qualities in the search tree. The search tree is depicted in Fig. 11. $Q$ values reflect the molecule's high-quality route expectation properly. The planner selects $M_1$ as the next molecule state node properly.

**Partial observation:** With a query cost of 0.01, the actor in ARP selects partial reaction qualities in the search tree to observe. The search tree is depicted in Fig. 12. It is observed that two reaction qualities are annotated. The molecule with the maximum $Q$ value maintains $M_1$. Nevertheless, the unannotated reaction quality of $R_2$ misdirects the $Q$ value estimate of $M_3$ to some extent. Though the ranking prior between

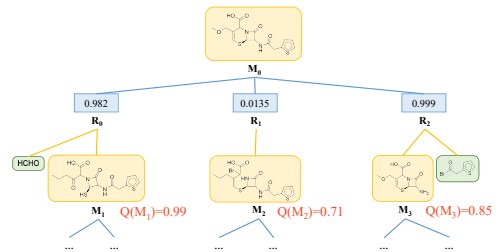

Figure 11: The search tree with query cost of 0.0

$M_2$ and $M_3$ changed compared with 11, the planner still selects $M_1$ to expand next. This phenomenon demonstrates the query ability of ARP to select the most impactful reactions to annotate qualities.

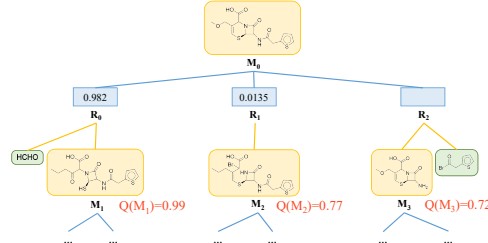

Figure 12: The search tree with query cost of 0.01

**None observation:** With a query cost of 0.05, the actor in ARP selects no reaction qualities in the search tree to observe. The search tree is depicted in Fig. 13. It is observed that the unannotated reaction qualities misdirect the $Q$ value estimates of three molecules. In contrast to Fig. 11 and Fig. 12, the next selected molecule changed into $M_2$. This issue, on the one hand, illustrates how reaction qualities benefit retrosynthetic planning, on the other hand, proves the active capability of utilizing the reaction quality annotations to find high-quality routes.

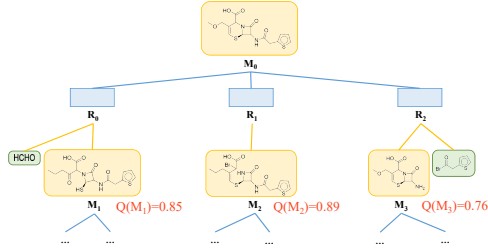

Figure 13: The search tree with query cost of 0.05

