# OpenReview forum: "Active Retrosynthetic Planning Aware of Route Quality"
_ICLR.cc/2024/Conference — ICLR 2024 poster_

### Official Review · Reviewer_23Kg · 2023-10-31

**Soundness:** 3 good
**Presentation:** 3 good
**Contribution:** 3 good
**Rating:** 6
**Confidence:** 4

**Summary:**

This work considers multi-step retrosynthetic search in a scenario where an expensive-but-accurate reaction quality metric is available. This motivates the authors to develop an algorithm that decides to query said metric adaptively, so that it can learn to trade-off the cost of querying it against other costs/rewards inherent to the routes themselves. The authors then run experiments to compare with existing retrosynthetic planning algorithms.

**Strengths:**

(S1): The framework proposed by the authors is original, well-motivated, and generally reasonable. The paper also includes valuable discussion.

- (a) In particular, the discussion on how routes produced by existing search algorithms may be not useful in the real world (due to e.g. the reaction being low yield or otherwise infeasible) is useful. This is something that should perhaps be given more attention in the ML-for-retrosynthesis  community.

- (b) The high-level modelling and architectural choices are reasonable and appropriate for the task at hand.

- (c) There are interesting connections of this work to retro-fallback [1] - a very recent parallel work - which is also concerned with the fact that the routes found by off-the-shelf algorithms may not be of high quality. However, I think the attack angles are somewhat complementary, as [1] tries to find _several_ routes that complement each other in their shortcomings, whereas this work focuses on _more accurately estimating the quality of a single route_ through a (possibly expensive) quality oracle.

(S2): The paper is generally well-written and easy to understand (although with some room for improvement; see the "Questions" and "Nitpicks" sections).

**Weaknesses:**

(W1): My main concern is that it is not very clear to me what the quality metric would actually be in a real-life scenario and how it would be implemented. First, the framework proposed by the authors seems to only make sense if querying the quality metric is _expensive but not too expensive_: if the quality oracle is not much more expensive that the single-step model itself (e.g. if it's a yield prediction model), then there is no need for the adaptive querying; if the metric is very expensive (e.g. asking a human chemist or running the reaction in the lab) then it wouldn't make sense to allow the algorithm to query it autonomously, even if it only does so for e.g. 20% of the cases (as that would still be prohibitively expensive and the human/lab would become a bottleneck). What would a realistic quality metric be? The authors in the paper use a trained model, which perhaps falls into the category of "not expensive enough to bother with adaptive querying", unless the model is *much* larger than the single-step model itself. Or perhaps the quality metric could result from e.g. running expensive quantum chemistry simulation, but this is a bit forward-looking as I don't think any existing frameworks for such calculations are robust enough to plug them in directly. Overall, at the very least I would expect more discussion on how the quality metric should be realized in practice.

(W2): The related works section is missing several references of established/SotA models and algorithms. On the single-step side, it is missing LocalRetro [2], RetroKNN [3] and RootAligned [4]. On the multi-step side, it is missing PDVN [5].

=== Summary ===

I like some aspects of this work, but the general real-world use case could be more discussed. I am looking forward to the rebuttal and, if productive, I may consider revising my score accordingly.

=== Other comments ===

(O1): The introduction section casually mentions that the average length of a retrosynthetic route is 3.36 without giving any sources. Where does this come from? I think this number would heavily depend on the precise assumptions, and for very challenging molecules the average route length would be even higher than this. It may be better to change this statement to be more vague (e.g. say that the routes are usually 3 or more steps) and give a citation, instead of using a very specific number which is only true if one fixes a lot of variables.



=== Nitpicks ===

Below I list nitpicks (e.g. typos, grammar errors), which did not have a significant impact on my review score, but it would be good to fix those to improve the paper further.

- "while this often leads to substantial query costs" -> in this context I would replace "while" with e.g. "but"

- in many places there is a missing space before parenthesis (see the "Multi-step planning" part of the "Related work" section for some examples)

- "with is initialized" -> "which"

- "Moreover, We define" -> lowercase "we"

- "There are three steps for each roll" -> "rollout"?

- First sentence in the "Reaction cost annotation" paragraph is a bit hard to parse

- "USPTI-MIT" -> "USPTO-MIT"?

- Equation 9 looks as if it's missing some text



=== References ===

[1] Tripp et al, "Retro-fallback: retrosynthetic planning in an uncertain world"

[2] Chen et al, "Deep Retrosynthetic Reaction Prediction using Local Reactivity and Global Attention"

[3] Xie et al, "Retrosynthesis Prediction with Local Template Retrieval"

[4] Zhong et al, "Root-aligned SMILES: a tight representation for chemical reaction prediction"

[5] Liu et al, "Retrosynthetic Planning with Dual Value Networks"

=== Update 21/11 ===

After the rebuttal, I increase my score to 6.

**Questions:**

(Q1): Could you expand on "critic takes both the current molecule and its preceding reaction cost as input"? Is the previous reaction cost literally concatenated into the molecule representation somewhere as a single floating point number?

(Q2): Is Equation 6 fully correct? I am confused about the fact that it does not include the discount factor $\gamma$, and also the way $N(s)$ is used seems slightly off; I may be wrong though.

(Q3): Could you elaborate on how the quality metric is used by the baseline algorithms like Retro* to produce the results from Table 2? I understand that for the baselines it is assumed the quality metric is always queried, which leads to 100% query rate. Could you discuss how the resulting quality value is used by the algorithms themselves during optimization?

(Q4): Is the depth of the exhaustive search used to normalize route quality the same as the maximum depth that the search algorithms are allowed to explore?

---

> ### Author Response · Authors · 2023-11-19
> **Response to Reviewer 23Kg(1/2)**
>
> We sincerely thank the reviewer for providing valuable feedback. We detail our response below point by point. Please feel free to let us know if you have any additional concerns or questions.
>
> **Real-life scenarios**
>
> > We wholeheartedly agree with the reviewer that the quality metric required by our framework in a real-life scenario should be **expensive but not prohibitively so**. This consideration constitutes the primary motivation behind our active planning framework, aiming to query a minimum number of reaction quality annotations while still planing high-quality routes.
> >
> > While our current implementation involves querying the surrogate model, our inspiration is drawn directly from real-life retrosynthesis planning scenarios, such as in online softwares like SYNTHIA, where **chemists are pivotal end users**. In this context, **integrating chemists as valuable resources into the AI planning process** will be invaluable for planning routes that are not only feasible but also of practical high-quality. We envision the successful deployment of our framework in this scenario for several reasons.
> > - Online annotation by chemists introduces minimal time delays and manageable labor costs, making it an **ideal candidate for a route quality metric** that is expensive but not prohibitively so.
> > - Our framework is intentionally designed to be **compatible with various types of annotations, including a coarse-grained quality rating from 0 to 10**. We believe such a rating is sufficient for the planner to make satisfactory decisions. Additionally, this rating can also be seamlessly integrated into our current framework by replacing the bucket index to which a quality value belongs (see details in Section 3.2 in the revision) with this discrete rating.
> >  - Chemists contribute **valuable insights beyond mere reaction yields**, such as knowledge about preferred reactions in real-world synthesis contexts, which can include factors like toxicity, material costs and work-up difficulty (post-process, like purification or separation).
>
>
> **Related works**
> > We thank the reviewer for pointing out these recent important related works [2-6], which we have cited and discussed in the revision version.
> > - **Related works on multi-step retrosynthesis planning**
> >   - [5] employs an RL-based method to improve single-step practicality, aiming to discover shorter routes with lower costs. However, **[5] does not consider real-world reaction qualities**, by treating not-dead reaction costs as 1 and evaluating routes based solely on their length.
> >   - As a complementary, our work contributes to finding high-quality routes by actively querying the real-world reaction qualities. We have proven the capability to reduce query costs while maintaining high route qualities. Importantly, **our active framework is compatible with retrosynthetic planning methods like [5]**.
> > - **Related works on single-step retrosynthesis prediction**
> >   - [2] addresses the identification and completion processes by employing local template prediction combined with a global attention mechanism.
> >   - [3] tackles the concern that fixed parameters might be sub-optimal by introducing a robust non-parametric local reaction template retrieval approach.
> >   - [4] resolves the issue of SMILES neglecting the characteristics of molecular graph topology and reaction atom transformations.
> >   **Our focus on retrosynthetic planning** can readily integrate with the success achieved by recent state-of-the-art single-step retrosynthesis prediction methods.
>
>
>
>
> **Nitpicks**
>
> > We appreciate your advice to help us improve this paper and revise the mentioned nitpicks in the revision.
>
> >[1] Binghong Chen et al. "Retro\*: Learning Retrosynthetic Planning with Neural Guided A\* Search".
> >
> >
> >[2] Chen et al, "Deep Retrosynthetic Reaction Prediction using Local Reactivity and Global Attention"
> >
> >[3] Xie et al, "Retrosynthesis Prediction with Local Template Retrieval"
> >
> >[4] Zhong et al, "Root-aligned SMILES: a tight representation for chemical reaction prediction"
> >
> >[5] Liu et al, "Retrosynthetic Planning with Dual Value Networks"

---

> ### Author Response · Authors · 2023-11-19
> **Response to Reviewer 23Kg(2/2)**
>
> **Q1**: Could you expand on "critic takes both the current molecule and its preceding reaction cost as input"? Is the previous reaction cost literally concatenated into the molecule representation somewhere as a single floating point number?
> > The preceding reaction cost is **concatenated in the format of its representation as $\mathcal{O}(a^r,a^q)$** in Eq.(2). Concretely,
> >   - When $a^q=1$, we derive $\mathcal{O}(a^r,a^q)$ from (1) discretizing continuous reaction quality values into $N^M$ discrete buckets, (2) learning $N^M$ trainable embeddings in $d^M$ dimensions for all buckets within our critic, and (3) determining the bucket index that the queried quality value $u$ belongs to and thereby the associated bucket embedding.
> >   - when $a^q=0$, $\mathcal{O}(a^r,a^q)=\mathbb{M}$ as a $d^M$-dimensional trainable embedding.
> > - In our implementation, we consider $d^M=512$ and $N^M=18$ buckets which are defined in Fig 6 in the revision. These buckets are obtained via (1) collecting ~28M reactions during planning by GRASP and Retro$^*$, (2) computing their reaction qualities by our surrogate model, and (3) defining the bucket boundaries to ensure that each bin covers a similar number of reactions.
>
> **Q2**: Is Equation 6 fully correct? I am confused about the fact that it does not include the discount factor $\gamma$, and also the way $N(s)$ is used seems slightly off; I may be wrong though.
>
> > We apologize for any confusion arising from the equations with respect to $Q^*$ and we further clarify Eq.(5) and Eq.(6) in the revision version.
> > - The visit count number $N(s)$ is used for the **reaction action selection**: $a^r_t=argmax_{a^{r}\in ch(s_t|T)}Q^*(s_t,a^{r})$ where $Q(s_t, a^r_t)$ is a value initialized by our critic network $Q_\theta(s_{t+1},\mathcal{O}(a^r_t,\pi_\phi(a^r_t)))$ and updated afterwards. The $Q^*$ function, quantifying the molecular synthesizability, is computed using the following equation:
> > $$
> > Q^*(s_t,a^r_t)=Q(s_t, a^r_t)+\beta\frac{\sqrt{N(s_{t-1},a^r_{t-1})}}{1+N(s_t,a^r_t)}
> > $$
> > - The visited count number $N(s)$ is **also employed in the update step**. The $Q(s_t, a^r_t)$ values are updated in association with the visit counts of the state-action pair using the following equation where $Q'$ is the $Q$ values of newly added state-action pairs. This update process also incorporates the discounting factor $\gamma$ as mentioned in the main text.
> > $$
> > Q'(s_t,a^r_t) = Q(s_t,a^r_t) + \frac{1}{N(s_t,a_t)}(\gamma Q'-Q(s_t,a^r_t))
> > $$
>
>
>
>
> **Q3**: Could you elaborate on how the quality metric is used by the baseline algorithms like Retro* to produce the results from Table 2? I understand that for the baselines it is assumed the quality metric is always queried, which leads to 100% query rate. Could you discuss how the resulting quality value is used by the algorithms themselves during optimization?
>
> > Retro* [1] leverages the A* search algorithm with the value of a molecule defined as
> > $$
> V_t(m|T)=g_t(m|T)+h_t(m|T).
> $$
> > - For the **heuristic function $h_t(m|T)$**, we retrained the value network $V_m$ with the same hyperparameters from [1].
> > - For the **reaction costs $g_t(m|T)$**,  we compute the reaction costs for each step **using our surrogate model** instead of the single-step probabilities.
> >
>
>
>
>
> **Q4**: Is the depth of the exhaustive search used to normalize route quality the same as the maximum depth that the search algorithms are allowed to explore?
>
> > Yes. We set the depth of both the exhaustive search and the planning process to be 6.
>
> > [1] Binghong Chen et al. "Retro\*: Learning Retrosynthetic Planning with Neural Guided A\* Search".

---

> ### Author Response · Authors · 2023-11-21
> **We would love to hear back from Reviewer 23Kg**
>
> Hi Reviewer 23Kg,
>
> We would like to follow up to see if our response addresses your concerns or if you have any further questions. We would really appreciate the opportunity to discuss this further if our response has not already addressed your concerns.
>
> Thank you again!

---

> > ### Comment · Reviewer_23Kg · 2023-11-21
> > **Response**
> >
> > > Online annotation by chemists introduces minimal time delays and manageable labor costs, making it an ideal candidate for a route quality metric that is expensive but not prohibitively so.
> >
> > I guess this makes sense; my thinking was that if the chemist is queried for e.g. 20% of the reactions then it would still be a bottleneck, but perhaps if it's a few % then that would be fine. A related question would be how to train the model to make use of the oracle without actually calling the oracle, as one may want to avoid introducing chemists during the training process to save their time (also, an untrained model may intially try to query the oracle much more often than is optimal). Perhaps it would be fine to train with a surrogate model of chemist preference and then use an actual chemist during inference.
> >
> > Either way, I think the direction outlined by the authors is interesting. It would be good to make sure the paper makes it clear that, in a real application, the quality metric may be "implemented" by querying a chemist.
> >
> > ---
> >
> > Other clarifications and changes (to e.g. the equations) look good to me. It's also good that the authors included more related work polled from the reviewers. I raise my score from 5 to 6.

---

> > > ### Author Response · Authors · 2023-11-22
> > > **Gratitude to Reviewer 23Kg**
> > >
> > > > I guess this makes sense; my thinking was that if the chemist is queried for e.g. 20% of the reactions then it would still be a bottleneck, but perhaps if it's a few % then that would be fine.
> > >
> > > We concur with the reviewer's assessment. Indeed, our proposed ARP framework has demonstrated its effectiveness in reducing the query rate (as low as **below $10$%**) when the query cost is high, all the while maintaining an almost consistent normalized route quality. Further details and evidence can be found in Table 3 and 4.
> > >
> > > > A related question would be how to train the model to make use of the oracle without actually calling the oracle, as one may want to avoid introducing chemists during the training process to save their time (also, an untrained model may intially try to query the oracle much more often than is optimal). Perhaps it would be fine to train with a surrogate model of chemist preference and then use an actual chemist during inference.  \
> > >
> > > We value the insightful suggestion provided by the reviewer, which is consistent with our current protocol.
> > > - Currently, training of our surrogate model involves two steps: (1) pre-training on the USPTO-MIT dataset, and (2) finetuning on an in-house expert dataset of routes featuring high-yield reactions. It is important to note that we introduce **step (2) deliberately to ensure that high predictive probabilities from our surrogate model align with high yields**. Thus, this surrogate model inherently reflects a preference towards high yields.
> > > - In the future, we plan to **extend the training of the surrogate model by incorporating chemist preference** gathered from the literature and in-house experiments. Additionally, during inference, we aim to adhere to our motivation by deploying the model through interactive queries with chemists.
> > > -------------------------------------
> > > We are grateful that the reviewer has acknowledged the resolution of other concerns and expressed satisfaction with our revision. We appreciate the reviewer's insightful feedback on the real-application scenario of our framework, and will provide further details on it in the Introduction of the final version.

---

### Official Review · Reviewer_zSxo · 2023-11-01

**Soundness:** 3 good
**Presentation:** 3 good
**Contribution:** 3 good
**Rating:** 8
**Confidence:** 4

**Summary:**

The paper addresses the challenge of retrosynthetic planning when the reaction cost is unknown during the planning process. To balance the trade-off between query costs and route quality evaluation, the authors introduce the Active Retrosynthetic Planning (ARP) framework. This framework employs an actor-critic model to decide whether to query reaction costs and to estimate molecule values. The ARP framework demonstrates a 6.2% improvement in route quality on a benchmark dataset and a 4.9% improvement on an annotated dataset, while reducing the number of query costs.

**Strengths:**

- The paper identifies a key issue in retrosynthetic planning: the assumption of knowing the reaction cost for every reaction during the planning phase is impractical. In response, the authors introduce a novel solution by leveraging an actor-critic framework from reinforcement learning to address this problem.
- The proposed method demonstrates strong empirical performance on top of state-of-the-art methods, as shown in comprehensive evaluations.
- The proposed method is generic, able to work alongside various existing retrosynthetic planners, enabling their capabilities to balance between query costs and route quality evaluation.
- The paper introduces the normalized route quality to normalize the scores for different synthesis routes, thus improving the accuracy of planning outcome evaluation.

**Weaknesses:**

- In the ablation study, actor+critic does not show consistent and significant improvement on success rate compared with random+critic baseline. The paper should include additional discussion to clarify this issue.

**Questions:**

- In the ablation study, why doesn't the actor+critic consistently and significantly improve the success rate compared to the random+critic baseline, as observed with normalized route quality?
- Why doesn't the success rate change monotonically with the query rate, unlike the normalized route quality?

---

> ### Author Response · Authors · 2023-11-19
> **Response to Reviewer zSxo**
>
> Thank you sincerely for your thoughtful and positive feedback on our work. We are particularly grateful for your recognition of the various aspects of our research. Below, we have provided a detailed explanation for your remaining concern as follows. Please do not hesitate to let us know if you have any further questions.
>
> **Q1**: In the ablation study, why doesn't the actor+critic consistently and significantly improve the success rate compared to the random+critic baseline, as observed with normalized route quality?
>
> **Q2**: Why doesn't the success rate change monotonically with the query rate, unlike the normalized route quality?
>
> > We address **Q1** and **Q2** together because of their high correlation. We have added a discussion into Section 4.2 and a case illustration in Appendix D in the revision.
> > - (1) **The success rate and route quality are two distinct and potentially conflicting objectives**, as illustrated in the case study of Fig 8 in Appendix D. In the figure, the root molecule $M_0$ has three candidate reactions, and $R_0$ is identified as a high quality reaction. However, the child molecule $M_1$ of $R_0$ is an unexpandable dead node with no further reaction candidates.
> >     - If the planner selects based only on observable next-state molecular structures of $M_1$, $M_2$ and $M_3$, **being agnostic of the reaction quality values of $R_0$, $R_1$ and $R_2$,** it might select $R_2$ for its most synthesizable next-state molecule $M_3$.
> >     - Alternatively, **given observable reaction quality values $R_0$, $R_1$ and $R_2$**, the planner could be misled into selecting $R_0$ due to its highest route quality.
> > - (2) The "actor" in Table 3/4 is a policy network $\pi_\phi$ that decides whether to query the quality of reactions. By taking the reaction quality as rewards (see Eq.(1) for details), **the actor is trained to prioritize querying those reactions that contribute to high-quality routes**.
> > - Considering (1) the disparity between high-quality and high success rates and (2) the actor optimized towards high-quality only, it is reasonable that
> >    - actor+critic does not show consistent and significant improvement on success rates compared to random+critic;
> >    - the success rate does not change monotonically with the query rate.

---

> > ### Comment · Reviewer_zSxo · 2023-11-21
> >
> > Thank you for addressing my concerns. I have no follow up questions.

---

> > > ### Author Response · Authors · 2023-11-22
> > > **Gratitude to Reviewer zSxo**
> > >
> > > Thank you for your prompt response and for acknowledging the addressed concerns. We sincerely appreciate your time and positive feedback on our work.

---

### Official Review · Reviewer_BDYr · 2023-11-04

**Soundness:** 3 good
**Presentation:** 2 fair
**Contribution:** 3 good
**Rating:** 6
**Confidence:** 3

**Summary:**

This paper aims to consider the route quality (i.e., yield of a reaction) in the retrosynthetic planning problem. Since querying such information requires labor-intensive lab/expert verification, this paper proposes an active planning framework to strike a balance between query cost and route quality. This framework consists of an actor that decides whether to query the cost of a reaction or not and a critic that evaluates whether to expand a molecule or not.

**Strengths:**

1. The route quality in terms of yield is of pivotal significance for practical CASP, and this paper is well motivated from this perspective;
2. The active planning framework is designed to consider route quality, with balancing the query cost; the actor-critic technique adopted in general is reasonable and justified;
3. The evaluation using defined metrics support the major claims of balancing query quality and rate.

**Weaknesses:**

1. The paper representation can be improved with better clarification. See details in Questions.
2. Some claims that motivate the method should be verified. See details in Questions.
3. Important related work [1] that targets on multi-step planning should be discussed.

[1] Liu, Songtao, et al. "FusionRetro: molecule representation fusion via in-context learning for retrosynthetic planning." International Conference on Machine Learning. PMLR, 2023.

**Questions:**

#### Unverified Claims
1. In reaction cost annotation, the authors adopt a surrogate model to provide reaction cost annotation, and hope “the model prioritizes the identification of high-yield reactions over high-frequency reactions”. Can the authors empirically verify this claim?
2. Are there real cases when high-yield reactions and high-frequency reactions are not overlapping? Intuitively, high-frequency reactions are usually high-yield ones, otherwise they cannot be frequently collected in a dataset, right?

#### Method
3. How is the state encoder $\mathcal{E}$ trained? It should also wrap $s_t$ in Eq. (4), right?
4. In Selection, is there any exploration or possibility to generate multiple routes?
5. How is the masked value M set?
6. During inference, does the method require the external surrogate model to obtain the reaction cost? How accurate/reliable is this surrogate model?

#### Writing
7. The term “reaction cost” is misleading and inconsistent: “cost” usually refers to some undesired property that should be as low as possible, however, this paper defines cost as yield (i.e., higher “cost” indicates better quality). This is misleading and counterintuitive. Meanwhile, “observing in Fig, 2 that a molecule with a low preceding reaction cost should be prioritized to expand first”, is clearly inconsistent with Fig. 2 (i.e., low-cost=low-quality route should not be prioritized).
8. The description of Expansion in the inference stage is not clear. Q* sometimes refers to a value, or a function (i.e., in Eq. (5)), which is confusing. What is Q*(a) and Q*(s) exactly?

---

> ### Author Response · Authors · 2023-11-19
> **Response to Reviewer BDYr(1/3)**
>
> We sincerely appreciate your constructive comments on this paper. We detail our response below point by point. Please kindly let us know if our response addresses the issues you raised in this paper.
>
> **Related works**
> >We thank the reviewer for pointing out this recent  and important related work [1], which we have duly cited and discussed in the revision.
> > - [1] introduces a novel multi-step planning approach via in-context learning, departing from conventional search algorithms. However, its primary objective is still evaluating success rates of planned routes.
> > - Our major contributions lie in an active framework capable of synsthesizing routes with not only high success rates but also high quality. Thus, we have proved the **compatibility and orthogonality of ours with other multi-step planning strategies** like Retro*; it is also readily extendable to state-of-the-art planning methods like [1].
>
> **Unverified claims**
>
> **Q1**: In reaction cost annotation, the authors adopt a surrogate model to provide reaction cost annotation, and hope “the model prioritizes the identification of high-yield reactions over high-frequency reactions”. Can the authors empirically verify this claim?
>
> > - Our active planning framework utilizes the **reaction quality** annotated by our surrogate model as **rewards**. Consequently, **as long as reaction qualities predicted by the surrogate model show high correlations with yields**, the model learns a policy that prioritizes high-yield reactions over high-frequency ones.
> > - We proceed to validate that reaction qualities by our surrogate model indeed demonstrates a high correlation with yields.
> >    - The training of our surrogate model involves two steps: (1) pre-training on the USPTO-MIT dataset, and (2) finetuning on an in-house expert dataset of routes featuring high-yield reactions. It is important to note that we **introduce step (2) precisely to ensure that high predictive probabilities from our surrogate model align with high yields**.
> >    - To evaluate our surrogate model, we resort to a route-with-yield test set. Following the method described in [2], we extract synthesis routes with yields from the USPTO-milligram-scale reaction yield dataset [3]. For evaluation purposes, we randomly select 200 routes, encompassing approximately 1000 reactions.
> >    - We thereby calculate the Pearson correlation coefficient(PCC) between the reaction quality predicted by our surrogate model and the literature yield. In Fig 7 of the revised manuscript, (a) illustrates the 0.059 PCC of the pre-trained model while (b) shows the **0.611 PCC of the finetuned model**, providing strong evidence that **the surrogate model accurately predicts yields**.
>
> **Q2**: Are there real cases when high-yield reactions and high-frequency reactions are not overlapping? Intuitively, high-frequency reactions are usually high-yield ones, otherwise they cannot be frequently collected in a dataset, right?
>
> > Frequently collected reactions in a single-step dataset are **not necessarily** high-yield, which we substantiate based on an analysis from [3] that explores yields reported in the open-source USPTO dataset.
> >
> > - The **USPTO dataset with reaction yields** in sub-gram scale [3] contains a large number of reactions and a broad range of superclasses, and a reaction distribution closely resembling that of the USPTO single-step dataset, such as USPTO-MIT.
> > -  The actual reaction yield distribution of the above dataset, originally presented in [3], is depicted in Fig 5c. Notably, a significant proportion of reactions within the dataset exhibits relatively low yields, affirming that the USPTO single-step dataset is **not inherently biased to high-yield reactions**.
> > - **High-frequency $\neq$ high-yield** Fig 5a (originally presented in [3]) shows various superclasses of reactions, where each color corresponds to a superclass and the coverage area of each color roughly represents the frequency of that superclass of reactions in the dataset. Combining Fig 5a and 5b, we conclude that high-frequency superclasses do not show a significant correlation with high yields. For example, the superclasses annotated in purple and cyan demonstrate low yields, with only the green reaction superclass corresponding to high yields in Fig 5b.
>
> > [1] Liu, Songtao, *et al*. "FusionRetro: molecule representation fusion via in-context learning for retrosynthetic planning".
> >
> > [2] Binghong Chen *et al*. "Retro\*: Learning Retrosynthetic Planning with Neural Guided A\* Search".
> >
> > [3] Philippe Schwaller *et al*. "Prediction of chemical reaction yields using deep learning"

---

> ### Author Response · Authors · 2023-11-19
> **Response to Reviewer BDYr(2/3)**
>
> **Method**
>
> **Q3**: How is the state encoder $\mathcal{E}$ trained? It should also wrap $\mathcal{E}$ in Eq. (4), right?
>
> > We first apologize for any ambiguity. We have updated Eq.(4) along with its descriptions in response.
> >  - If $\mathcal{E}$ is an **offline estimator** such as the value estimator in Retro*, it is trained before implementation in our framework. The parameters of $\mathcal{E}$ are frozen and not evolved in training of the critic.
> >  - If $\mathcal{E}$ is an **online estimator**, it can be regarded as part of the critic parameters and is wrapped with the other critic parameters  for training.
>
> **Q4**: In Selection, is there any exploration or possibility to generate multiple routes?
>
> > We thank the reviewer's great comment, and have accordingly updated Section 3.3 to detail the definition of **$Q^*$ which involves exploration**.
> >
> > Concretely, the function $Q^*$ is defined as
> > $$
> > Q^*(s_t,a^r_t)=Q(s_t, a^r_t)+\beta\frac{\sqrt{N(s_{t-1},a^r_{t-1})}}{1+N(s_t,a^r_t)}
> > $$
> > which follows the UCT(Upper Confidence Bound applied to Trees) function to balance between exploitation of the route with the maximum $Q$ and exploration of those less frequently visited routes.
> >
>
> **Q5**: How is the masked value M set?
>
> > - We set the mask as a $d^M$-dimensional trainable embedding.
> > - Meantime, it is crucial to emphasize the **consistency in the representation of $\mathcal{O}(a^r,a^q)$**:
> >   - When $a^q=1$, we derive $\mathcal{O}(a^r,a^q)$ from (1) discretizing continuous reaction quality values into $N^M$ discrete buckets, (2) learning $N^M$ trainable embeddings in $d^M$ dimensions for all buckets within our critic, and (3) determining the bucket index that the queried quality value $u$ belongs to and thereby the associated bucket embedding.
> >   - when $a^q=0$, as mentioned above, $\mathcal{O}(a^r,a^q)=\mathbb{M}$ as a $d^M$ dimensional embedding.
> > - In our implementation, we consider $d^M=512$ and $N^M=18$ buckets which are defined in Fig 6 in the revision. These buckets are obtained via (1) collecting ~28M reactions during planning by GRASP and Retro$^*$, (2) computing their reaction qualities by our surrogate model, and (3) defining the bucket boundaries to ensure that each bin covers a similar number of reactions.
>
> **Q6**: During inference, does the method require the external surrogate model to obtain the reaction cost? How accurate/reliable is this surrogate model?
>
> > - Our major contributions revolve around exactly an **active** planning framework that trains a querying actor/critic during training and later applies it **during inference** to plan a high-quality route with **minimal reaction quality annotation from an external surrogate model or a chemist expert**.
> > - Still, ours is compatible with existing multi-step planning methods such as Retro* as backbone planners. In scenarios where an external surrogate model or a chemist is unavailable, ours reduces to the backbone planner.
> > - In **the response to Q1**, we have empirically validated the effectiveness of our surrogate model. Besides, our **case study in Appendix E** showcases that ours achieves **much higher route quality ratings by chemists**. This outcome provides strong evidence that the surrogate model faithfully reflects yields.

---

> ### Author Response · Authors · 2023-11-19
> **Response to Reviewer BDYr(3/3)**
>
> **Writing**
>
> **Q7**: The term “reaction cost” is misleading and inconsistent: “cost” usually refers to some undesired property that should be as low as possible, however, this paper defines cost as yield (i.e., higher “cost” indicates better quality). This is misleading and counterintuitive. Meanwhile, “observing in Fig, 2 that a molecule with a low preceding reaction cost should be prioritized to expand first”, is clearly inconsistent with Fig. 2 (i.e., low-cost=low-quality route should not be prioritized).
>
> > We apologize for any confusion arising from the term "reaction cost". For a more straightforward and intuitive understanding, we have revised all instances of the term "reaction cost" into "reaction quality" as well as Fig 2 in the updated version.
>
> **Q8**: The description of Expansion in the inference stage is not clear. $Q^*$ sometimes refers to a value, or a function (i.e., in Eq. (5)), which is confusing. What is $Q^*(a)$ and $Q^*(s)$ exactly?
>
> > For a better understanding, we have replaced $Q^*(s_t)$ and $Q^*(a^r_t)$ with a unified form $Q^*(s_t, a^r_t)$ in Eq 5 and Eq 6. $Q^*$ is a function that quantifies the synthesizability of a state-action pair as outlined below. Inside the equation, $N(s, a^r)$ denotes the visit count number. $Q(s_t, a^r_t)$ is a value initialized by $Q_\theta(s_{t+1},\mathcal{O}(a^r_t,\pi_\phi(a^r_t)))$ and updated afterwards.
> > $$
> > Q^*(s_t,a^r_t)=Q(s_t, a^r_t)+\beta\frac{\sqrt{N(s_{t-1},a^r_{t-1})}}{1+N(s_t,a^r_t)}
> > $$
> > During a selection step, We recursively perform a reaction action selection $a^r_t=argmax_{a^{r}\in ch(s_t|T)}Q^*(s_t,a^{r})$ and obtain the next state $s_{t+1}$ by tranition function $\mathcal{P}$ until reaching a leaf molecule node. In a update step, we perform a $Q(s_t,a^r_t)$ value backward traversal as follow where $Q'$ is the $Q$ values of newly added state-action pairs.
> > $$
> > Q'(s_t,a^r_t) = Q(s_t,a^r_t) + \frac{1}{N(s_t,a_t)}(\gamma Q'-Q(s_t,a^r_t))
> > $$

---

> ### Author Response · Authors · 2023-11-21
> **We would love to hear back from Reviewer BDYr**
>
> Hi Reviewer BDYr,
>
> We would like to follow up to see if our response addresses your concerns or if you have any further questions. We would really appreciate the opportunity to discuss this further if our response has not already addressed your concerns.
>
> Thank you again!

---

> ### Comment · Reviewer_BDYr · 2023-11-21
> **Thanks for the Explanations**
>
> I appreciate the authors providing detailed evidence and explanations to my questions. I have also read the updated version with improved clarity. Since my concerns were all addressed, I raised my score from 5 to 6.

---

> > ### Author Response · Authors · 2023-11-22
> > **Gratitude to Reviewer BDYr**
> >
> > We are pleased that the concerns raised by the reviewer have been successfully addressed. Once again, we extend our gratitude for the time and effort the reviewer dedicated to thoroughly reviewing our paper and providing valuable feedback, which has significantly contributed to enhancing the current version of our work.

---

### Official Review · Reviewer_iEQU · 2023-11-22

**Soundness:** 3 good
**Presentation:** 2 fair
**Contribution:** 2 fair
**Rating:** 6
**Confidence:** 5

**Summary:**

an RL based search algorithm that incorporates some definition of route quality for retrosynthesis is proposed.


edit after discussion
- I am still not 100% convinced about the novelty of this work, but nevertheless will raise the score.

**Strengths:**

- interesting new ideas of using RL for scoring applied to retrosynthesis

**Weaknesses:**

- somewhat disconnected from previous evaluation
- the model to assign quality is not described in enough detail
- it is not convincingly defined what reaction quality is
- prior work is not correctly attributed

"Practical efficacy: we, for the first time, draw
an insight into the disappointing practicality of existing retrosynthetic planners that regard single-step probabilities as reaction qualities."

Different and flexible ways to incorporate step and route cost beyond single step probabilities) was already included in many prior works, for example Segler (2018), Coley (2019) and Schwaller (2020). These works also used reaction feasibility models, which are a proxy for quality. Please reference this accordingly.
also, the method seems to be very similar to Liu et al 2023, which is also a very flexible framework able to incorporate arbitary cost. this needs to be discussed more carefully.
I think the paper can receive higher score if prior work is referenced properly, but in the current form this is insufficient.

**Questions:**

- I would suggest to order the related work section, in particular the Multi-step planning section, by the order in which the works appeared, because subsequent works influenced each other.

**Details Of Ethics Concerns:**

-

---

> ### Author Response · Authors · 2023-11-23
> **Response to Reviewer iEQU(1/3)**
>
> We thank the reviewer for providing feedback. Below, we detail our response point by point.
>
> **Clarification on Our Major Contributions**
> > In response to the reviewer's summary characterizing our work as ``an RL based search algorithm that incorporates some definition of route quality for retrosynthesis is proposed'', we express concerns about potential misinterpretations of the major contributions of our work. To address this, we elaborate as follows.
> > - Our contribution lies in not first incorporation of route quality related metrics into retrosynthesis.
> > - Our contribution lies in being the first in acknowledging the cost of annotating reaction quality, and thereby enabling the practical incorporation of costly-yet-accurate route quality evaluation into retrosynthesis through an **active and interactive** manner. Our motivations and contributions can be encapsulated in the following points:
> >   - (1) There has been **a lack of efficient-to-evaluate and meantime qualified route quality metrics**; for example,
> >     - (a) Predicted yield by a prediction model [efficient but inaccurate]: constructing accurate and robust yield prediction models has historically been challenging, explaining existing models being confined to specific reaction types.
> >     - (b) Route length [efficient but unfaithful]: The illustration in Fig. 1 of the main text demonstrates that a short route might yield less than a longer one.
> >     - (c ) Single-step probabilities [efficient but inaccurate]: As noted in our Introduction, single-step probabilities primarily reflect feasibility rather than reaction quality.
> >     - (d) SCScore [efficient but inaccurate]: While SCScore quantifies synthesizability (feasibility), it neglects actual reaction performance quality.
> >     - (e) Chemist rating [more accurate but less inefficient]: Chemists offer an overall evaluation of reaction yield, but this process is less efficient than a prediction model.
> >     - (f) Lab experiment verification [accurate but inefficient]: While reflective of groundtruth reaction quality, including yields, this approach is exceedingly time-consuming.
> >   - (2) The preceding works, including [1]-[4] the reviewer mentioned, operate **under the assumption that annotating route quality is labor-free and trivial**. Consequently, they accommodate metrics (a)-(d), as incorporation of highly accurate evaluation (e)-(f) for every reaction along the planned route is deemed prohibitively costly.
> >   - (3) Thus, we are motivated to propose this **"active"** retrosynthetic planning framework, strategically **minimizing the reactions where we query metrics such as (e)-(f) while sufficiently improve the route quality**.
> >      - Example: In the case of a molecule requiring 25 synthesis steps, our framework endeavors to pinpoint only 2 key reactions (10% querying rate) for the evaluation of either (e) or (f).
> >      - This framework **maintains compatibility with existing retrosynthesis planning methods**, a validation substantiated in our experiments on Retro* and GRASP. Take Liu et al. 2023 [3] as an example. If training the PDVN with ARP online, the synthesizability value Network $R$ and the cost value network $Q$ in PDVN are incorporated into the estimator $\mathcal{E(s_t)}$ in the ARP framework by $\mathcal{E(s_t)}=R(s_{t-1},a^r_{t-1})*Q(s_{t-1},a^r_{t-1})+(1-R(s_{t-1},a^r_{t-1}))*c_{dead}$.
> >   - (4) In our current implementation, we adopt the surrogate model as a surrogate for (e)-(f), which however does not compromise the effectiveness of our active planner.
>
> > [1] Segler, M., Preuss, M. & Waller, M. Planning chemical syntheses with deep neural networks and symbolic AI. Nature 555, 604–610 (2018).
> >
> > [2] John S. Schreck, Connor W. Coley, and Kyle J. M. Bishop Schreck. Learning Retrosynthetic Planning through Simulated Experience. ACS Central Science 2019
> >
> > [3] Guoqing Liu, Di Xue, Shufang Xie, Yingce Xia, Austin Tripp, Krzysztof Maziarz, Marwin Segler, Tao Qin, Zongzhang Zhang, and Tie-Yan Liu. 2023. Retrosynthetic planning with dual value networks. In Proceedings of the 40th International Conference on Machine Learning (ICML'23)
> >
> > [4] Schwaller P, Petraglia R, Zullo V, Nair VH, Haeuselmann RA, Pisoni R, et al. Predicting Retrosynthetic Pathways Using a Combined Linguistic Model and Hyper-Graph Exploration Strategy. ChemRxiv. Cambridge: Cambridge Open Engage; 2019.

---

> ### Author Response · Authors · 2023-11-23
> **Response to Reviewer iEQU(2/3)**
>
> **Weakness**
> **W1**: Somewhat disconnected from previous evaluation
> > - To begin, we seek to draw a comparison between the metrics employed in our framework and those utilized in prior studies.
> >   - Our metrics: (c1) success rate, (o2) query rate, and (o3) normalized route quality.
> >   - Metrics in prior studies: (c1) success rate, (p2) average route length, and (p3) number of single-step model calls.
> > - We have already substantiated the superior performance of our framework in terms of (c1) success rate compared to previous planning methods.
> > - Concerning (p2) average route length, its application in previous works aimed to quantify a specific aspect of route quality. However, it is worth noting that this metric is **independent of real-world route quality**, as illustrated in Figure 1 of the main text. This motivates our adoption of (o3) normalized route quality for a more faithful evaluation.
> > - we extend our assessment to (p3) the number of single-step model calls. The performances of various baselines and our framework are outlined below. The GRASP approach achieves the fewest iterations among all baselines. Our approach, GRASP with ARP, surpasses GRASP by 1.2 in the benchmark dataset and by 3.9 in the expert dataset. In conclusion, our framework **enhances the planning efficiency** of the implemented planners."
> >
> >
> > | Model | Retro*-0 | Retro* | Retro*-0+ | Retro*+ | EG-MCTS-0 | EG-MCTS | GRASP | Retro* with ARP | GRASP with ARP |
> > | ---- | ---- | ---- | ---- | ---- | ----- | ---- | ---- | ----- | ----- |
> > | Benchmark test set | 69.2 | 53.5 | 52.6 | 48.7 | 45.9 | 43.1 | 41.3 | 46.0 | 40.1 |
> > | Expert test set | 36.7 | 33.2 | 28.9 | 25.7 | 33.4 | 28.9 | 22.6 | 25.6 | 18.7 |
>
> **W2**: The model to assign quality is not described in enough detail
>
> > - We had elucidated the training methodology for acquiring the surrogate model that assigns quality in the **paragraph of Reaction quality annotation in Section 4.1**.
> >   - "*Initially, the method in Guo et al. (2020) is employed to pre-train a model utilizing the USPTO-MIT dataset, followed by the fine-tuning of the model in reactions derived from the high-quality, expert-annotated dataset*"
> >   - We introduced this fine-tuning step on the expert dataset consisting of high yield reactions precisely to ensure that high predictive probabilities from our surrogate model **align with high yields**.
> > - We had included additional information on the implementation details in Appendix C and presented a thorough evaluation of the surrogate model itself in Appendix D.
>
> **W3**: It is not convincingly defined what reaction quality is
>
> > - We had defined the labor-intensive reaction quality in the Introduction section.
> >   - **Line 4 Section 1**: "*The ideal reaction qualities that meet real-world chemical practicability, e.g., **yield of a reaction**, can be either annotated experimentally in a laboratory or by experienced chemists.*"
> > - We had also detailed how to implement the reaction qualities so far for our framework in the **paragraph of Reaction quality annotation in Section 4.1**:
> >    - "*Unfortunately, there is no established large-scale available reaction data set with respective reaction qualities or promising reaction performance, e.g. yield, prediction model(Jiang et al. (2022)), and both lab verification and expert annotations are expensive and time-consuming. We adopt a surrogate model to provide reaction quality annotations. ... During deployment, it is practical to replace the surrogate model with a chemist to provide online annotations, e.g. a coarse-grained quality rating from 0 to 10.*"

---

> ### Author Response · Authors · 2023-11-23
> **Response to Reviewer iEQU(3/3)**
>
> **W4**: Prior work is not correctly attributed, regarding Segler (2018) [1], Coley (2019) [2], Schwaller (2020) [4] and Liu et al 2023 [3].
>
> > As clarified in the aforementioned ``**Clarification on Our Major Contributions**'', our primary contributions stand distinct from the prior works [1-4]. Furthermore, our proposed framework further is compatible with and enhances the real-world practicality of there existing planners.
> >   - The planning methods in [1]-[4] only accommodate reaction cost metrics that are **efficient to have yet inaccurate to reflect real-world reaction quality such as yields**.
> >     - [1] utilzes a fixed heuristic based on starting material prices and reaction labor costs, unrelated to yields.
> >     - [2][3] consider the cost for each reaction as a uniform 1 and optimize towards shortest routes. However, a short route might have a lower yield than a long route.
> >     - [4] relies on a forward prediction likelihood biased to the high-frequency reactions and the SCScore. SCScore, which gauges molecule synthesizability (a.k.a. feasibility) but overlooks the real-world reaction performance quality.
> >  - The methodologies above **require the annotation of costs for every reaction along a route**. In the pursuit of incorporating reliable reaction qualities derived from chemists or lab experimentation, which entail significant costs, these methods tend to be economically impractical for real-world deployment.
> > - Therefore, our essential contribution lies in introducing  a framework designed to  **(1) actively decide when to query human evaluation (surrogate model in our current implementation)** and **(2) select a minimum number of reactions that are most informative to observe their quality annotations**. Our framework not only reduces query costs but also renders eminently practical in real-world scenarios.
> > - By specifically addressing the challenge of annotation costs, a facet often overlooked by previous works, our framework **not only accommodates but synergistically integrates with existing models**. This compatibility facilitates the seamless implementation of their models within our framework, thereby extending their active query capabilities.
>
> **Questions**
>
> **Q1**: I would suggest to order the related work section, in particular the Multi-step planning section, by the order in which the works appeared, because subsequent works influenced each other.
>
> > We arrange the order of the related work section by the search algorithm previous studies used. We highlight the search algorithm categories, e.g. $A^*$, $MCTS$，$RL$, in the revision version updated.
>
>
> >[1] Segler, M., Preuss, M. & Waller, M. Planning chemical syntheses with deep neural networks and symbolic AI. Nature 555, 604–610 (2018).
> >
> >[2] John S. Schreck, Connor W. Coley, and Kyle J. M. Bishop Schreck. Learning Retrosynthetic Planning through Simulated Experience. ACS Central Science 2019
> >
> >[3] Guoqing Liu, Di Xue, Shufang Xie, Yingce Xia, Austin Tripp, Krzysztof Maziarz, Marwin Segler, Tao Qin, Zongzhang Zhang, and Tie-Yan Liu. 2023. Retrosynthetic planning with dual value networks. In Proceedings of the 40th International Conference on Machine Learning (ICML'23)
> >
> >[4] Schwaller P, Petraglia R, Zullo V, Nair VH, Haeuselmann RA, Pisoni R, et al. Predicting Retrosynthetic Pathways Using a Combined Linguistic Model and Hyper-Graph Exploration Strategy. ChemRxiv. Cambridge: Cambridge Open Engage; 2019.

---

### Author Response · Authors · 2023-11-20
**General response**

Dear Reviewers:

We thank the Reviewers for their diligent efforts and high-quality reviews. We have revised our paper to incorporate the valuable feedback provided in the reviews. For your convenience, we have temporarily highlighted these updates in red. Please find below a summary of the changes we made:
>
> **Additional Experiments**:
> - We provide an additional experiment in Appendix C to demonstrate a significant correlation between our surrogate model and reaction yields. (Reviewer BDYr Q1)
>
> **Clarification**:
> - We have revised all the term “reaction cost” into “reaction quality” as well as Fig 2 and Fig 3 for a more straightforward understanding.(Reviewer BDYr Q7)
> - An analysis in Appendix A of the issue that high-frequency reactions do not overlap with high-yield ones.
> - An illustration in Appendix B of a data binning strategy to discretize the continuous reaction quality values into discrete buckets. (Reviewer BDYr Q5 and Reviewer 23Kg Q1)
> - An analysis in Section 4.2 and a case study in Appendix D of the inconsistency between the success rate and route quality optimization objectives. (Reviewer zSxo Q1 and Q2)
> - The clarification in Section 3.3 of Eq 5 and Eq 6 which tie up the selection, expansion and update steps with a $Q^*(s_t, a^r_t)$ function during the inference. (Reviewer BDYr Q8 and Reviewer 23Kg Q2)
> - The clarification in Section 3.3 of exploring the less frequently visited routes by a UCT function.
> - An analysis in Appendix E of a real-world scenario of our method that incorporates human-in-the-loop to provide annotations.(Reviewer 23Kg)
>
>
> **Writting Errors**:
>
> We have corrected the nitpicks listed by Reviewer 23Kg .
>
> **Related works discussion**:
>
> We cite and discuss [1-6] in the revision version.
>
>[1] Liu, Songtao, et al. “FusionRetro: molecule representation fusion via in-context learning for retrosynthetic planning.”
>
>[2] Tripp et al, “Retro-fallback: retrosynthetic planning in an uncertain world”
>
>[3] Chen et al, “Deep Retrosynthetic Reaction Prediction using Local Reactivity and Global Attention”
>
>[4] Xie et al, “Retrosynthesis Prediction with Local Template Retrieval”
>
>[5] Zhong et al, “Root-aligned SMILES: a tight representation for chemical reaction prediction”
>
>[6] Liu et al, “Retrosynthetic Planning with Dual Value Networks”

We sincerely hope that our response and revisions have addressed all concerns raised by the reviewers. Please let us know if our response satisfactorily answers the questions you had for this paper. Thank you once again for your time and effort.

Best regards,

Authors

---

### Meta-Review · Area_Chair_b5fy · 2023-12-06

**Metareview:**

The paper introduces an Active Retrosynthetic Planning (ARP) framework, employing an actor-critic model to address the challenge of unknown reaction costs in retrosynthetic planning. The framework aims to balance query costs with route quality evaluation, demonstrating improved route quality and reduced query costs in evaluations.

Key strengths (+) and weakness (-) are summarized as below:

+ Novelty and Practical Relevance: The approach to handling unknown reaction costs in retrosynthetic planning is innovative and addresses a real-world issue.
+ Empirical Performance: The ARP framework shows strong performance, improving upon existing methods in evaluations.
+ Versatility: The framework is compatible with various existing retrosynthetic planners.

- Clarity: Some reviewers noted a need for more clarity in model descriptions and the definition of reaction quality.
- Prior Work: There's a consensus that the paper could better attribute existing related work, particularly in multi-step planning and route quality considerations.
- Ablation study: Inconsistencies in the actor critic model’s performance in the ablation study

**Justification For Why Not Higher Score:**

The paper did not receive a higher score primarily due to issues with clarity in presentation and insufficient attribution to prior work. The concerns raised in the ablation study also suggest a need for further analysis to substantiate the claimed improvements, impacting the perceived robustness of the findings.

**Justification For Why Not Lower Score:**

The paper's innovative approach, strong empirical performance, and practical relevance in a real-world context justify the current rating. The ARP framework's compatibility with existing planners and its potential impact in the field of retrosynthetic planning are significant.

---

### Decision · Program_Chairs · 2024-01-16

Accept (poster)